# Integration of head and body orientations in the macaque superior temporal sulcus is stronger for upright bodies

Yordanka Zafirova[1,2], Rufin Vogels[1,2]*

[1]Laboratorium voor Neuro- en Psychofysiologie, Department of Neurosciences, Leuven, Belgium; [2]Leuven Brain Institute, Leuven, Belgium

## eLife Assessment

This **important** study examines the neuronal mechanisms underlying visual perception of integrated face and body cues. The innovative paradigm, which employs monkey avatars in combination with electrophysiological recordings from fMRI-defined brain areas, provides **compelling** evidence on face and body integration. These results should be of wide interest to system and cognitive neuroscientists, psychologists, and behavioural biologists working on visual and social cognition.

*For correspondence:
rufin.vogels@kuleuven.be

## Abstract

The neural processing of faces and bodies is often studied separately, despite their natural integration in perception. Unlike prior research on the neural selectivity for either head or body orientation, we investigated their interaction in macaque superior temporal sulcus (STS) using a monkey avatar with diverse head–body orientation angles. STS neurons showed selectivity for specific combinations of head–body orientations. Anterior STS (aSTS) neurons enabled more reliable decoding of head–body configuration angles compared to middle STS neurons. Decoding accuracy in aSTS was lowest for head–body angle pairs differing only in sign (e.g. head–body orientation difference of ±90° relative to the anatomical midline), and highest for aligned (0°) head–body orientations versus those with maximum angular difference. Inverted bodies showed diminished decoding of head–body orientation angle compared to upright bodies. These findings show that aSTS integrates head and body orientation cues, revealing configuration-specific neural mechanisms, and advance our understanding of social perception.

## Introduction

Faces and bodies are typically studied separately, although both are parts of the same agent (*Freiwald et al., 2016*; *Tsao and Livingstone, 2008*; *de Gelder et al., 2010*; *Peelen and Downing, 2007*). Despite the distinct research traditions, behavioral and fMRI studies in humans suggest that face and body processing interact (*Hu et al., 2020*; *Taubert et al., 2022*). Furthermore, face-selective units in macaque mid superior temporal sulcus (STS) face-selective patches respond to an object occluding a face on top of a body (*Arcaro et al., 2020*), which agrees with monkey (*Fisher and Freiwald, 2015*) and human fMRI (*Cox et al., 2004*) studies that showed that a blurred head on top of a body can activate face-selective regions. Recently, we showed that anterior inferior temporal (IT) neurons respond stronger to face–body configurations in which the face is located at its natural position on top of the body than to unnatural face–body configurations, showing that the face and body interact in a configuration-specific manner in IT (*Zafirova et al., 2024*; *Zafirova et al., 2022*). While these studies demonstrate that face and body processing interact in the visual temporal cortex, the extent and underlying mechanisms of this interaction remain unexplored.

The responses of most face-selective STS neurons are modulated by the 3D orientation of the head (*Freiwald and Tsao, 2010*; *Perrett et al., 1992*; *Perrett et al., 1984*; *Perrett et al., 1985*). Likewise, the responses of most body-selective STS neurons are sensitive to body orientation (*Bao et al., 2020*; *Kumar et al., 2019*). This might not be surprising because both head and body orientation are important social cues, for example for the attention of a viewed agent or the direction of an agent's action. In these studies, either face or body orientation, but not both, were manipulated. Here we address the outstanding question of whether the orientation of the head and body interact in the STS: does the selectivity for head orientation depend on the body orientation, and vice versa? Are STS neurons selective for a particular combination of head and body orientation, suggesting a selectivity for both head and body features in a particular pose, for example a selective response for a frontal body with a head turned to the right? Our study was also motivated by human psychophysical studies that showed that body and head orientation interact in social perception (*Hietanen, 2002*; *Moors et al., 2015*; *Vrancken et al., 2017*), although it is unclear at which level – sensory or higher – the observed interactions occurred.

Here, we manipulated independently the head and body orientation of a monkey avatar. We employed both anatomically possible and impossible head–body orientations (e.g. a frontal body with a backward-oriented head, i.e. 180° difference), allowing us to assess whether anatomically possible and impossible head–body orientation angles are encoded differently by STS neurons.

Behavioral studies in humans (*Griffin and Oswald, 2022*) and monkeys (*Matsuno and Fujita, 2018*) provided evidence for greater body discrimination for upright compared to inverted bodies. This body-inversion effect has been interpreted as providing evidence for the holistic encoding of bodies, similar to that proposed for faces. Some studies suggested that the body-inversion effect is driven by the head (for review, see *Griffin and Oswald, 2022*). However, other studies have observed a behavioral body-inversion effect for headless bodies (for a meta-analysis, see *Griffin and Oswald, 2022*). So far, no neural correlates of the body-inversion effect for whole bodies have been described at the single-unit level. Here, we compared the responses and body–head interaction of STS units between upright and inverted bodies, assessing whether these units show stronger responses, or higher body selectivity and body–head interactions for upright bodies.

We recorded the responses of units at two posterior–anterior levels of the ventral STS, in and surrounding patches that were activated more by images of monkeys compared to objects in an fMRI study with the same subjects (*Zafirova et al., 2022*; *Figure 1A*). Because it has been proposed that face–body interactions are more prominent in anterior IT (*Fisher and Freiwald, 2015*; *Hu et al., 2020*; *Zafirova et al., 2022*), we expected stronger head–body orientation interactions in the anterior STS (aSTS).

## Results

In experiment 1, we targeted two fMRI-defined patches that were defined by a higher activation to images of monkeys compared to objects (*Zafirova et al., 2022*; *Figure 1A*). The posterior patch was located in the ventral bank of the mid STS (mSTS) and the anterior one was about 12 mm more anterior in the ventral bank of the STS (aSTS). In experiment 1, we recorded well-isolated single units in both regions. In the Selectivity test (Materials and methods), we compared the responses to a set of monkeys, faces, headless bodies, and objects (see *Zafirova et al., 2022*). We took neurons (Materials and methods) for further examination when their mean net response to the monkey, face, or body was at least twice the mean response to the corresponding object control condition. Thus, we obtained 98 aSTS and 100 mSTS neurons, split approximately equally across the two subjects. In both regions, the averaged responses of the selected neurons were strongest for the monkey images, followed closely by the body and the face, and much weaker for the objects (*Figure 1B*). Some neurons responded stronger to faces than bodies, or vice versa, and showed strong selectivity for different body or face images (*Figure 1B*).

These neurons were tested in the Head–body Orientation test in which we manipulated independently the orientation of the head and the body of a monkey avatar (Materials and methods; *Figure 2A*). The monkey avatar was either sitting (P1) or standing (P2). For each pose, we had 64 conditions in which the orientation of the head and body was manipulated independently in steps of 45° (*Figure 2A*). To separate the effect of head/body location from head or body orientation, we presented all images at two locations: monkey-centered (MC) and head-centered (HC; *Figure 2B*).

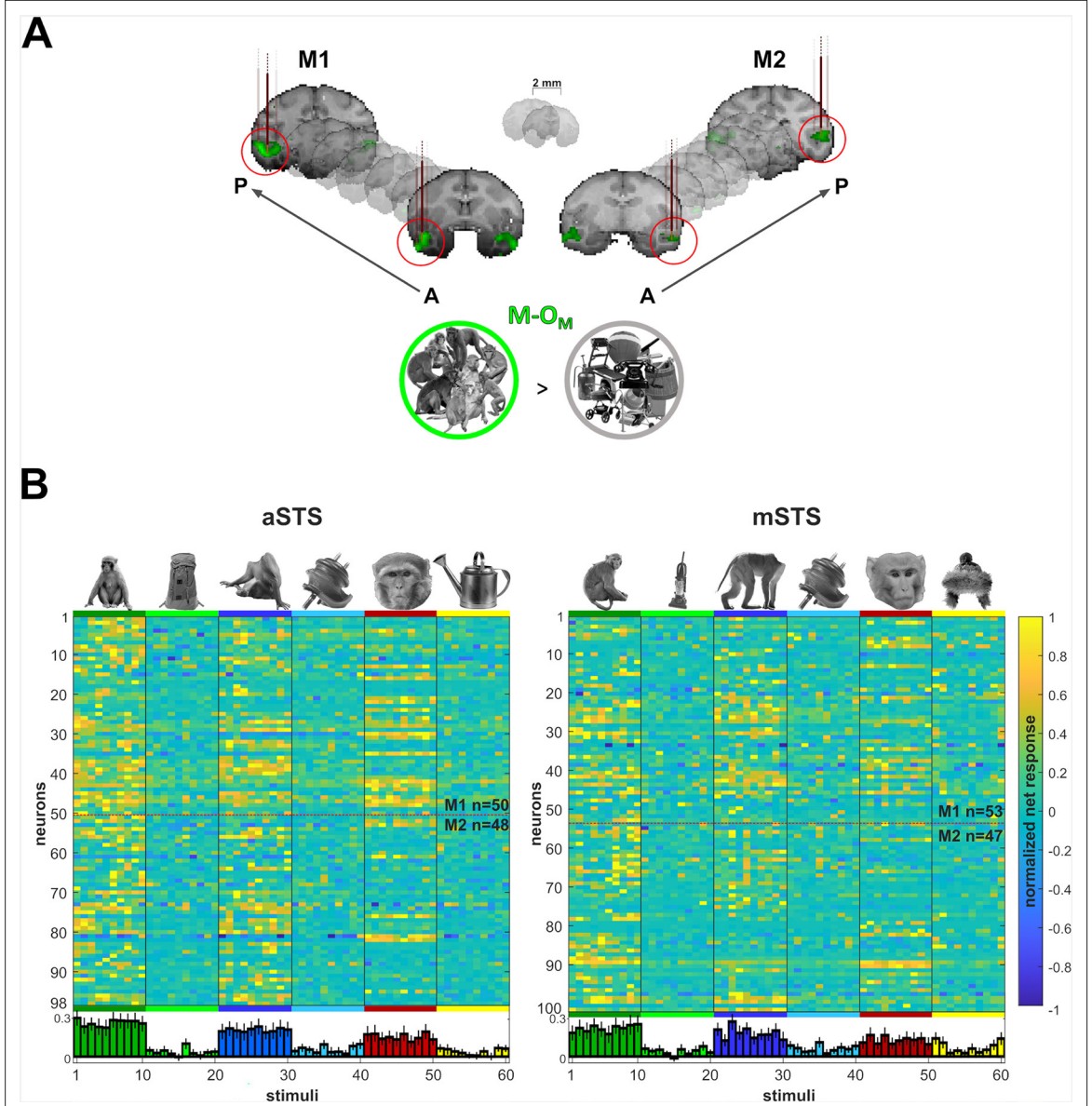

**Figure 1.** Recording locations and stimulus selectivity. (**A**) Targeted patches (red circles) shown in the native space of M1 (left) and M2 (right) on 2 mm coronal slices. Brown vertical lines illustrate penetrations. Monkey patches were defined by the contrast monkeys > monkey control objects (M-O$_M$; p < 0.05 FEW). (**B**) Normalized net responses in the Selectivity test (experiment 1) in anterior superior temporal sulcus (aSTS) (left) and mid superior temporal sulcus (mSTS) (right). Rows represent neurons from M1 and M2 (separated by a dotted line). Columns correspond to the stimuli, color-coded per category with a bar and one example image. Below, bars in corresponding colors show the mean normalized net response to each stimulus (95% confidence intervals (1000 resamplings)). *n* indicates the number of neurons.

## Responses for a monkey are on average less than the summed responses for the body and head (experiment 1)

For each pose, the head and body of the frontal, back, and two lateral orientations were also presented in isolation at the same locations as in the avatar. The latter was done for head–body configurations of zero and straight (0° and 180°) angles between the head and body orientation (*Figure 2B, C*). This allowed us to compare the responses to the isolated body and head with those to the monkeys having the same head and body at the same locations. For each case for which there was a significant excitatory response to either the monkey or the corresponding head and body compared to the baseline, we computed the monkey-sum index (MSI) contrasting the net response to the monkey and the sum of the net responses to its corresponding head and body (Materials and methods). MSI larger

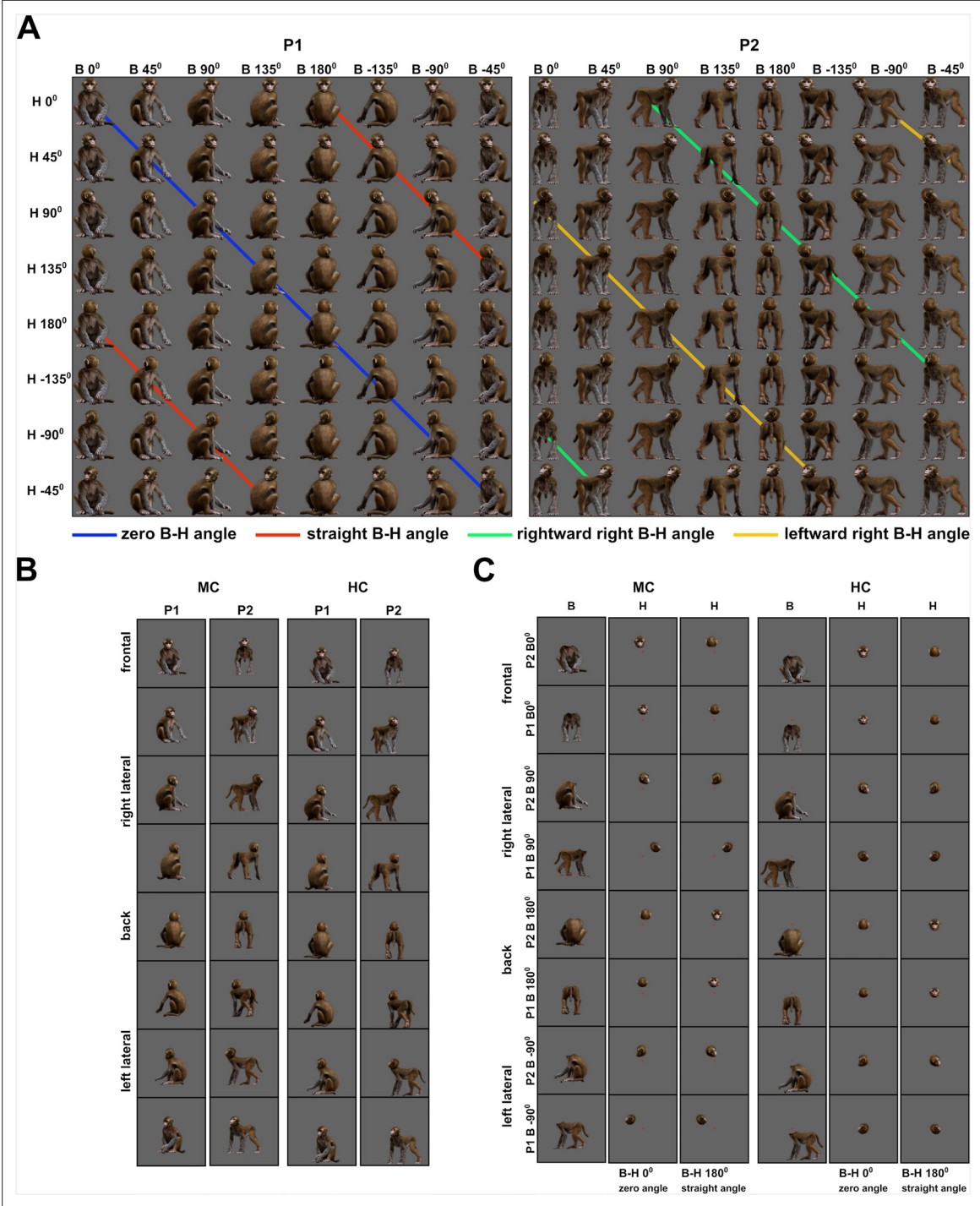

**Figure 2.** Monkey avatar stimuli for the Head–body Orientation test (experiment 1). (**A**) Sixty-four combinations of head and body orientations for the sitting (P1, left) and the standing pose (P2, right). Body orientation varies by row and head orientation varies by column. Different colors group the eight avatar orientations of four head–body angles. (**B**) Example stimuli of P1 and P2 in the monkey-centered (MC, left) and the head-centered (HC, right) conditions, with zero angle between head and body. (**C**) Headless bodies (B) and heads (H) for the zero and straight angle head–body configurations, showing the frontal, back, and two lateral views for P1 and P2. Images are shown for MC (left) and HC (right) conditions and were positioned to match their corresponding avatars in (**A**). The location of the red fixation target is indicated in (**B**) and (**C**).

than 0 corresponds to a superadditive response to the monkey while MSI smaller than 0 indicates a subadditive response. For both subjects and regions, the median MSIs for P1 and P2 were smaller than 0 (*Figure 3A*), which agrees with a single-unit study in anterior IT that employed images of real macaques (*Zafirova et al., 2024*). Surprisingly, we observed neurons that responded to either the headless body or head but did not respond to the whole monkey that included the same body and head (MSI close to –1; *Figure 3A*). The weak responses to the monkey were not limited to cases in which the head–body configuration was anatomically impossible, that is a straight angle between the head and the body. In fact, cases in which MSI was between –0.9 and –1 included 53% and 41% of natural head–body configurations in aSTS and mSTS, respectively. In line with a previous study (*Zafirova et al., 2024*), only a minority of neurons demonstrated strong superadditive responses to the monkey (MSI between 0.9 and 1: 4.3% and 6.2% in aSTS and mSTS, respectively). *Figure 3E* illustrates the responses to heads, bodies, and monkeys of example units with different values of MSI. These data show that heads and bodies interact at the level of the STS.

## Interaction of head and body orientation tuning (experiment 1)

The main question of this study was whether head and body *orientation tuning* interact. Thus, we performed a two-way ANOVA on the responses (Materials and methods) to the 64 head–body orientation conditions, with factors the orientation of the head and body (*Figure 2A*). The large majority of neurons showed a significant main effect (p < 0.05) of body orientation followed by a main effect of head orientation (*Figure 3B*). Importantly, a large fraction of neurons showed a significant interaction between head and body orientation. This was true for both centerings (MC and HC) and poses (P1 and P2) in each region (aSTS and mSTS, *Figure 3B*).

These results do not necessarily provide evidence for genuine head and body orientation interactions because orientation changes could be confounded with location changes of the monkey part within a centering condition (*Figure 2B*). To address this, we first assessed whether the tuning for body/head orientation was location tolerant by correlating the responses to the 64 head–body configurations between MC and HC. This was done for all neuron and pose combinations in which there was a significant response in either centering condition (split-plot ANOVA). The median correlations were 0.68 and 0.57 in aSTS and mSTS, respectively (*Figure 3C*), indicating position tolerance of the orientation selectivity in both regions. In line with its higher hierarchical stage and thus expected higher location tolerance, aSTS showed significantly higher correlations than mSTS (Wilcoxon rank-sum test, p = 0.008; tested after averaging the correlation coefficients of the poses per neuron; $n$ = 75 aSTS and $n$ = 77 mSTS neurons). The higher correlations for aSTS were not a result of differences in response reliability between the two regions: the median Spearman–Brown corrected split-half Pearson correlation (Materials and methods) was 0.82 and 0.81 for the MC and HC aSTS neurons ($n$ = 131 cases) and 0.85 and 0.89 for the MC and HC mSTS neurons ($n$ = 126 cases). The relatively higher reliability for mSTS shows that the lower correlation of the head–body orientation selectivity between the two centerings for mSTS neurons is not due to lower reliability. Furthermore, correcting the correlations for reliability showed a lower median corrected correlation for mSTS (median: 0.72) compared to aSTS (0.90).

To assess the position tolerance of the head–body orientation interaction per se, we computed how many neurons for each pose show a significant interaction (two-way ANOVA; see above) of head and body orientation for both centerings. In aSTS, 24% of the neurons responding for both centerings of a pose ($n$ = 135) showed a significant interaction in both centerings, and this proportion was similar for mSTS (26%; $n$ = 127; *Figure 3B*). Notably, for neuron and pose combinations with significant head–body interactions in both centerings, the correlations between responses to the 64 head–body orientation conditions were similar to those observed in the whole population (*Figure 3C*).

Next, we isolated the interaction of head and body orientation by computing the residuals of the responses from a linear additive model in which the responses to a head–body configuration are an addition of the mean responses to the body (averaging across the eight head orientations) and head (averaging across the eight body orientations) orientations. The residuals were computed for both centerings and then correlated across the centerings per neuron and pose combination. We computed these correlations for those cases in which the two-way ANOVA (see above) showed a significant interaction for both centerings. The median correlation between the residuals of the two centerings was 0.23 (quartiles: 0.08–0.36; $n$ = 33) for mSTS, whereas it was 0.41 (0.21–0.55; $n$ = 33; median larger than 0: p = 9.35e−07) for aSTS, which was significantly greater than for mSTS (Wilcoxon

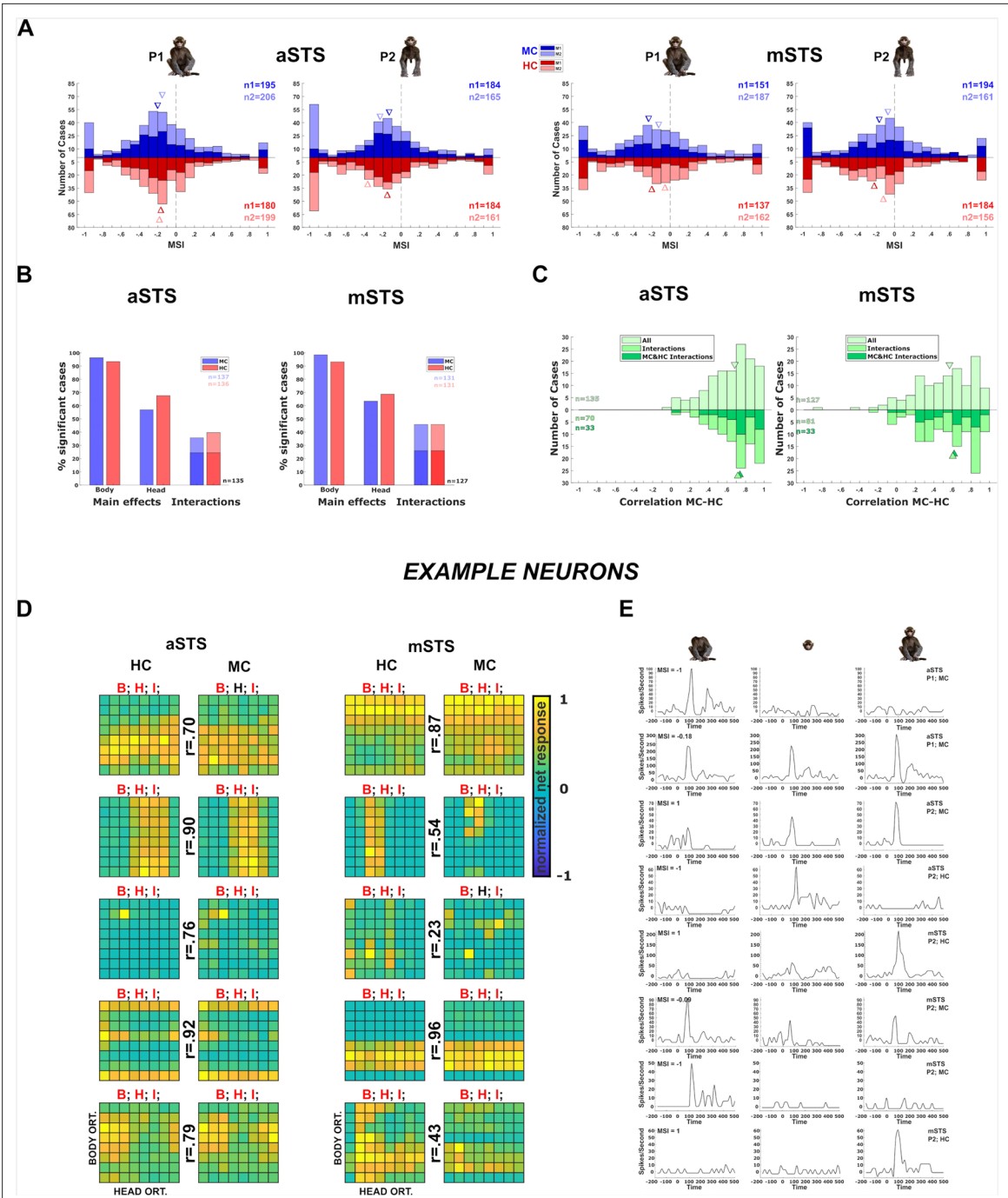

**Figure 3.** Neuronal responses in the Head–body Orientation test (experiment 1). (**A**) Distribution of monkey-sum index (MSI) computed for the responses for all orientation, pose (P1 and P2), and centering (monkey-centered (MC) (blue) and head-centered (HC) (red)) combinations in anterior superior temporal sulcus (aSTS) (upper panel) and mid superior temporal sulcus (mSTS) (lower panel). Data for M1 and M2, and medians (triangles) are shown in different hues (n1 = number of cases for M1; n2 = number of cases for M2). (**B**) Significant main effects and interactions of the factors body and head orientation, for P1 and P2 in MC (blue) and HC (red) conditions in corresponding colors in aSTS (left) and mSTS (right). Pooled data of both subjects. Significant interactions in both MC and HC are highlighted with darker blue and red (number of cases (n) in black). (**C**) Distribution of the correlation coefficients between responses to the head–body configurations in MC and HC conditions in aSTS (left) and mSTS (right). Data from both subjects and poses were pooled. Correlation coefficients for all cases are shown above the x-axis. Below the x-axis, the light green bars represent the correlation coefficients for cases with a significant interaction (either MC or HC), while the dark green bars indicate the coefficients for cases with a significant interaction in both MC and HC. N and medians (triangles) are shown in corresponding colors. (**D**) Normalized net responses of example neurons to the 64 combinations of head and body orientations in aSTS (left) and mSTS (right) for MC and HC conditions. Body orientation (ORT) varies

*Figure 3 continued on next page*

*Figure 3 continued*

by row and head orientation varies by column, from 0° to 315°. Main effects and interactions of the factors body and head orientation B, H, and I are shown for each neuron in each centering (significant ones are in red, see (**B**) for details), as well as the correlation coefficients *r* between responses to the head–body configurations in MC and HC conditions. (**E**) Net responses of example neurons to bodies, heads, and the corresponding monkey configurations with MSI of 1, –1, or the median. Data for P1 and P2, MC and HC, and aSTS and mSTS (smoothed with the Matlab function *interp1*).

The online version of this article includes the following figure supplement(s) for figure 3:

**Figure supplement 1.** Correlation between head- and body orientation selectivity between the sitting (P1) and standing pose (P2; experiment 1).

rank-sum test; p = 0.008). These data demonstrate a position tolerance of the interaction of head and body orientation in aSTS, but less so in mSTS.

Single neurons showed a variety of head–body orientation tunings (*Figure 3D*). Interestingly, some neurons (e.g. *Figure 3D*) were tuned to a particular combination of a head and body orientation irrespective of centering (e.g. the third aSTS example neuron of *Figure 3D*). The tuning for head–body orientation generalized across the two poses for the population of mSTS and aSTS neurons that responded significantly to each pose (*Figure 3—figure supplement 1*).

## Decoding of orientation-invariant head–body configuration (experiment 1)

Given the head–body orientation interactions in the STS, we examined whether we could decode head–body configurations with different angles between head and body in an orientation-invariant way from our sample of neurons using a linear classifier (linear kernel support vector machine (SVM)). We trained the classifier to distinguish two angles after pooling the responses of all eight orientations of a head–body orientation angle configuration (Materials and methods; examples in *Figure 4A*). Because the training used responses to all orientations per angle and because the same head and body orientations were present in both classes, we forced the classifier to employ the *combination* of head and body orientation. We used eightfold cross-validation with training and test trials evenly distributed across orientations (Materials and methods). We will report below the classification accuracies obtained when training was performed using trials of one centering condition (MC) and testing for the other centering condition (HC), requiring generalization across positions. The classification accuracies for training on the HC and testing on the MC condition were highly similar (*Figure 4—figure supplement 1*). We employed for each region 60 neurons, 30 randomly selected for each monkey subject, as that was the maximum number of neurons with responses in both centering conditions (MC and HC) per subject and recording location (aSTS and mSTS). The classification was performed using raw firing rates. We report the mean and standard deviation, across 100 resamplings, of the classification accuracies for the test trials.

*Figure 4B* shows the mean classification accuracies for pairs of head–body orientation angles, plotted for each of four reference angles (0° (zero angle), 180° (straight angle), 90° (rightward right angle), and –90° (leftward right angle)) versus other angles. For the zero and straight reference angles, the classification accuracy increased with the absolute difference between the angles, reaching a maximum of about 0.7 classification accuracy for zero versus straight angles. Classification accuracies were smaller for mSTS compared to aSTS. Surprisingly, for the rightward (90°) and leftward (–90°) right angles as references, classification scores were low (*Figure 4B*), which contrasts with the higher scores for the same 180° orientation difference when the angles were zero and straight (this difference was significant for both centerings and poses in aSTS; all p-values <0.01). Classification scores tended to be higher for 90° differences in angles (e.g. right vs. straight) than for the 180° difference (rightward vs. leftward right angles) for the rightward or leftward right reference angles.

The decoding accuracies showed the same pattern, although were higher, when training and testing were performed with the same centering, requiring no generalization across locations (*Figure 4—figure supplement 1*). A neuron-dropping analysis applied to the decoding of zero versus straight angles for the aSTS sample (MC training and HC testing) suggested that less than half of the neurons contributed positively to the decoding (*Figure 4—figure supplement 2*). Most but not all of the 10 'best' neurons, that is those that contributed most to the decoding, showed a significant interaction between head and body orientation. Furthermore, more of the 10 'best' neurons (at least 50% (95% confidence interval: 19–81%)) showed individually a significant effect of head–body orientation angle

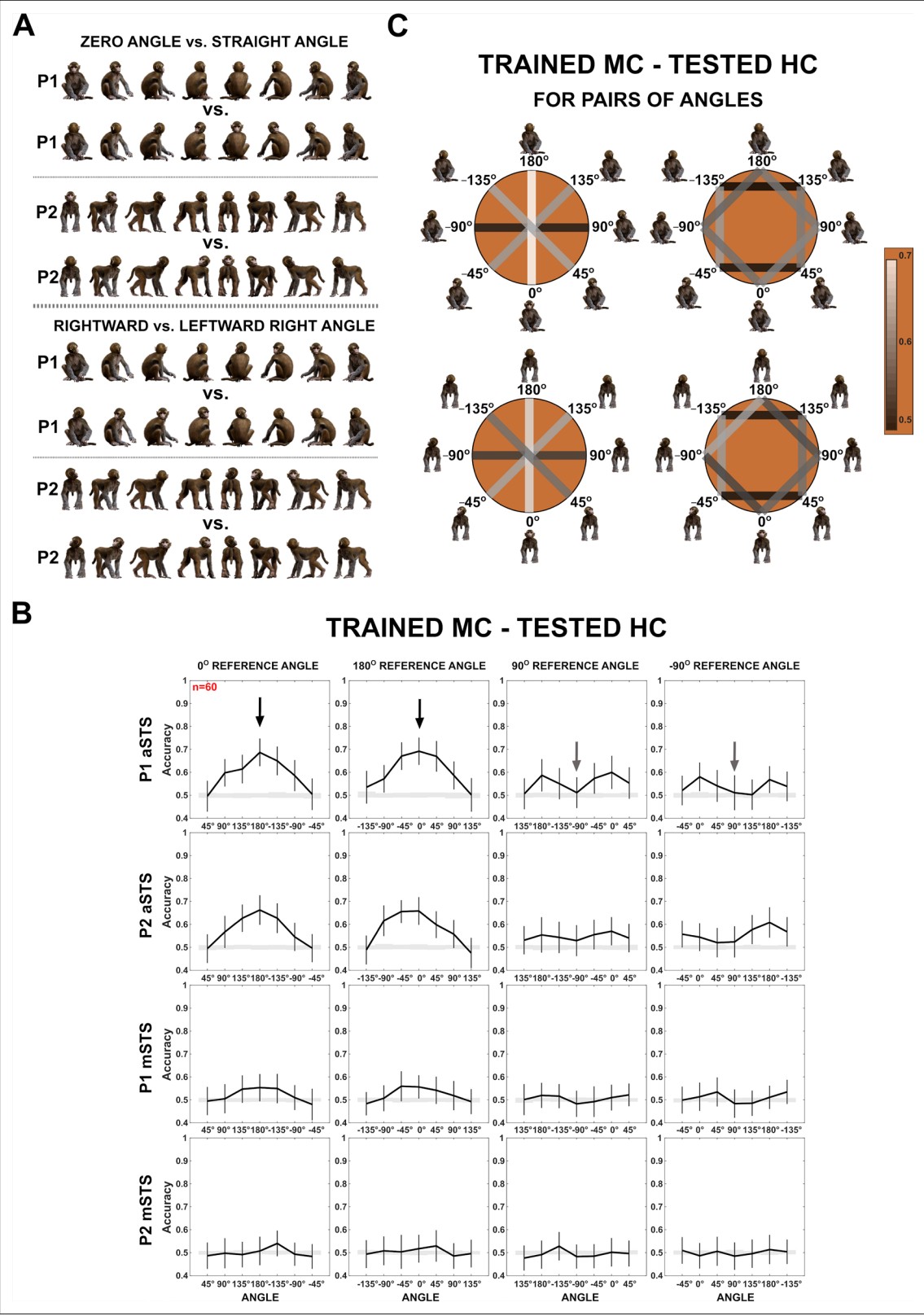

**Figure 4.** Decoding of head–body angles from neuronal responses in the Head–body Orientation test (experiment 1). (**A**) Head–body configurations with zero and straight angles (black arrows in **B**) and leftward and rightward right angles of head and body (gray arrows in **B**), of P1 and P2, at eight avatar orientations. (**B**) Mean decoding accuracies for the head–body configurations with zero, straight, leftward, and rightward right angles versus other head–body angles as indicated, in anterior superior temporal sulcus (aSTS) (top two panels) and mid superior temporal sulcus (mSTS) (bottom

*Figure 4 continued on next page*

*Figure 4 continued*

two panels), for poses P1 and P2. *N* indicates the number of neurons. Results are for monkey-centered (MC) training and head-centered (HC) testing. Responses to the eight orientations of a configuration (rows in **A**) were pooled when training the classifier. Error bars show standard deviations across resamplings. The shaded area indicates null distribution (stimulus labels permutation). (**C**) Decoding accuracies for different head–body angle pairs in aSTS, for P1 (top) and P2 (bottom). Decoded pairs are connected with lines, with decoding accuracies indicated by color. Training was performed after pooling the eight orientations of a head–body configuration defined by the angle of the head and body. Results are from MC training and HC testing. Avatars of one orientation of the head–body configuration angles are shown.

The online version of this article includes the following figure supplement(s) for figure 4:

**Figure supplement 1.** Decoding of head–body angles in the Head–body Orientation test (experiment 1).

**Figure supplement 2.** Neuron-dropping analysis (experiment 1).

**Figure supplement 3.** Decoding of head and body orientation (experiment 1).

on their responses (one-way ANOVA; $p < 0.05$) than expected for a random sample of the population of neurons (maximal expected percentage 16%; see *Figure 4—figure supplement 2* for more details). Nonetheless, the tuning profiles were diverse (*Figure 4—figure supplement 2*), suggesting population coding of head–body orientation angle.

We applied also a more stringent classification in which the trained and tested orientations differed: we trained the classifier for the frontal, back, and two lateral orientations (0°, 180°, 90°, and –90°, i.e. '+' orientations in *Figure 5A*), and tested with the oblique ones (45°, 135°, –135° and –45°, i.e. 'x'

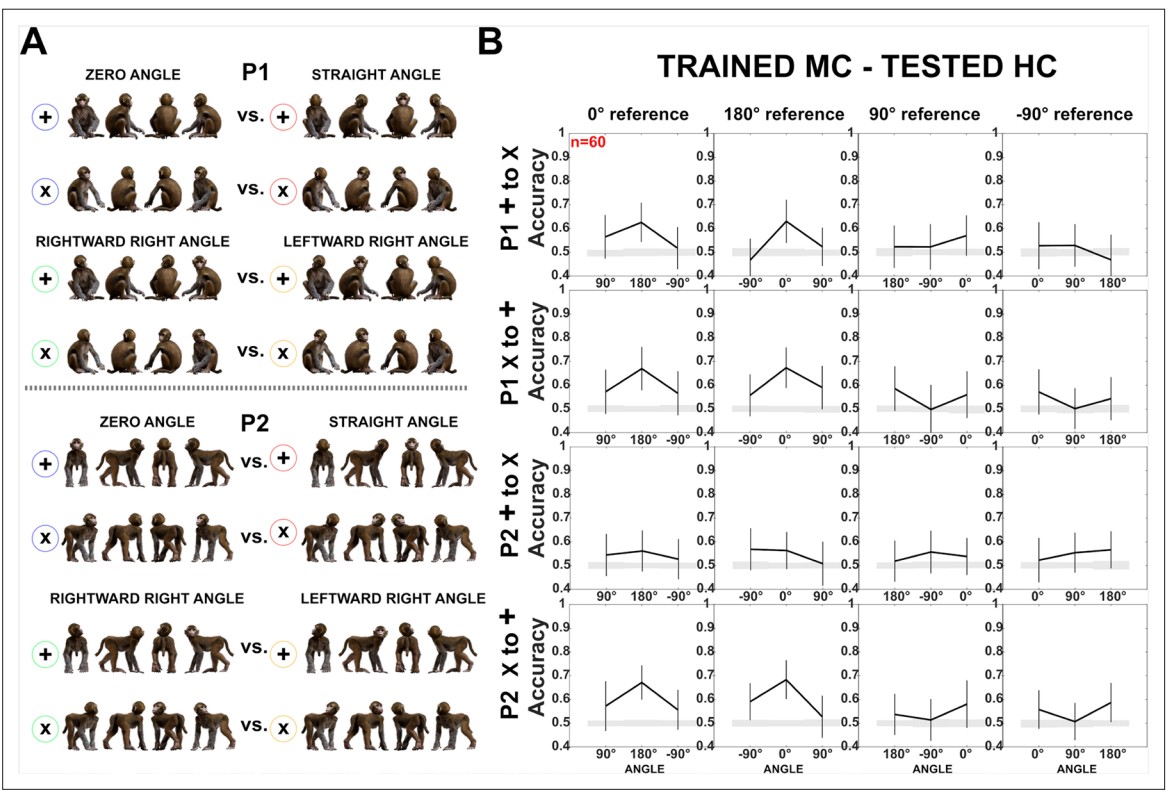

**Figure 5.** Decoding of head–body angles across orientations (experiment 1). (**A**) Head–body configurations with zero and straight angles, and leftward and rightward right angles for P1 and P2, split into four orientations for training and testing. (**B**) Decoding accuracies for the head–body configurations with zero, straight, leftward, and rightward right angles in anterior superior temporal sulcus (aSTS), for P1 (top) and P2 (bottom); *n* indicates the number of neurons. Results are for monkey-centered (MC) training and head-centered (HC) testing, using four orientations for training and testing. Panels 1 and 3: classifier trained on '+' orientations and tested on 'x' orientations; panels 2 and 4: classifier trained on 'x' orientations and tested on '+' orientations. Same conventions as *Figure 4*.

The online version of this article includes the following figure supplement(s) for figure 5:

**Figure supplement 1.** Decoding of head–body angles across orientations (experiment 1).

**Figure supplement 2.** Dissimilarity matrices of neural responses for head and body orientations (experiment 1).

orientations in *Figure 5A*), and vice versa (*Figure 5A*, *Figure 5—figure supplement 1*). Again, the trained and tested responses were from different centerings. For this analysis, we employed the aSTS sample, as that produced the highest decoding accuracies previously. For P1, the performance accuracies for the aSTS sample were well above chance for the zero versus straight angle decoding, and, as before, were lower for the leftward versus rightward right angle decoding (*Figure 5B*; for brevity, we will refer to these angles as left and right angles, respectively, below). The same was true for P2 for two of the four centering and orientation training–test combinations (*Figure 5B*; *Figure 5—figure supplement 1*). However, for the other two combinations, the performance accuracies for the zero versus straight angles were barely above chance and were at a similar level as those for the right versus left angle decoding (*Figure 5B*; *Figure 5—figure supplement 1*). This interaction between generalization across orientation and centering for P2 is likely related to the larger variations in the eccentricity of the head when changing the orientation and centering of the standing avatar (see *Figure 5—figure supplement 1*). Nonetheless, when generalization across orientation but not across centering was required, accuracies for P1 and P2 were similar for all decodings (for the zero vs. straight angle: classification accuracies for P1 ranging from 0.64 to 0.72 and for P2 ranging from 0.66 to 0.72; for the right vs. left angle decoding: P1 0.49–0.54 and P2 0.48–0.56).

The marked difference in accuracy between the zero versus straight angle decoding and the right versus left angle decoding in aSTS was not trivially due to differences in mean response strength since there was no consistent effect of head–body orientation angle on mean response strength across poses (Friedman ANOVA; all p-values for both poses and centerings >0.1).

The left and right angles have the same absolute angle between head and body (90°), only the sign of the angle differs with respect to the anatomical midline (90° vs. –90°). The confusion between these configurations suggests that this neuronal population is relatively insensitive to the sign of the angle. This was confirmed by other decodings (*Figure 4C*; *Figure 4—figure supplement 1*). We observed a close-to-chance decoding of 45° versus –45° angles and 135° versus –135° angles, unlike the better classification accuracy for the same 90° differences between angles of 0° and 90° or 180° and –90°, the latter but not the former differing in the absolute angle between head and body. Other configurations that differ in absolute angle also produced good decoding performance: 45° versus –135° and 135° versus –45°, better than the left versus right angles, which differed only in sign (*Figure 4C*; *Figure 4—figure supplement 1*).

One explanation of the poor decoding for the right versus left angles is that in those conditions either the body or the face is at a lateral orientation. Neurons in anterior IT are less sensitive to differences between mirror-symmetric images (*Freiwald and Tsao, 2010*; *Rollenhagen and Olson, 2000*). If such neurons contribute to the computation of the head–body orientation angle, poor decoding of right and left angles can be expected since these images are, to some extent, mirror-symmetric. Therefore, we assessed the presence of mirror-symmetric tuning for the head and/or body in the same sample of aSTS neurons used for the decoding. To assess this for the head, we first averaged the responses, normalized by the maximum responses across the 64 stimuli, for the 8 monkey images having the same head (but a different body) orientation and then computed the distance matrix using the $1 - $ Pearson $r$ distance metric. Similarly, to assess mirror-symmetric tuning for body orientation, we averaged the responses to the monkeys having the same body orientation. The distance matrices showed evidence for mirror-symmetric responses for the lateral head and body orientations for P2 and face orientations of P1 (*Figure 5—figure supplement 2*). Mirror symmetry was less the case for the body of P1, which fits the poorer image symmetry of the body in the case of the P1 lateral views. Thus, the lower sensitivity for mirror-symmetric heads or bodies may have contributed to the lower decoding of head–body angles that differed only in their sign. However, the difference in decoding accuracy between the zero versus straight angle and the left versus right angle, especially when training on the 'x' orientations and testing on '+' orientations (*Figure 5*), cannot be fully attributed to the mirror symmetry of the avatar images.

## Decoding head–body orientation angle from the summed responses to body and face versus from responses to the whole body (experiment 2)

We obtained a second dataset (experiment 2) from the same subjects in which we could test the main findings of experiment 1 for a larger sample of neurons. Additionally, we evaluated whether the angle between the head and the body could be computed from their individual orientations when

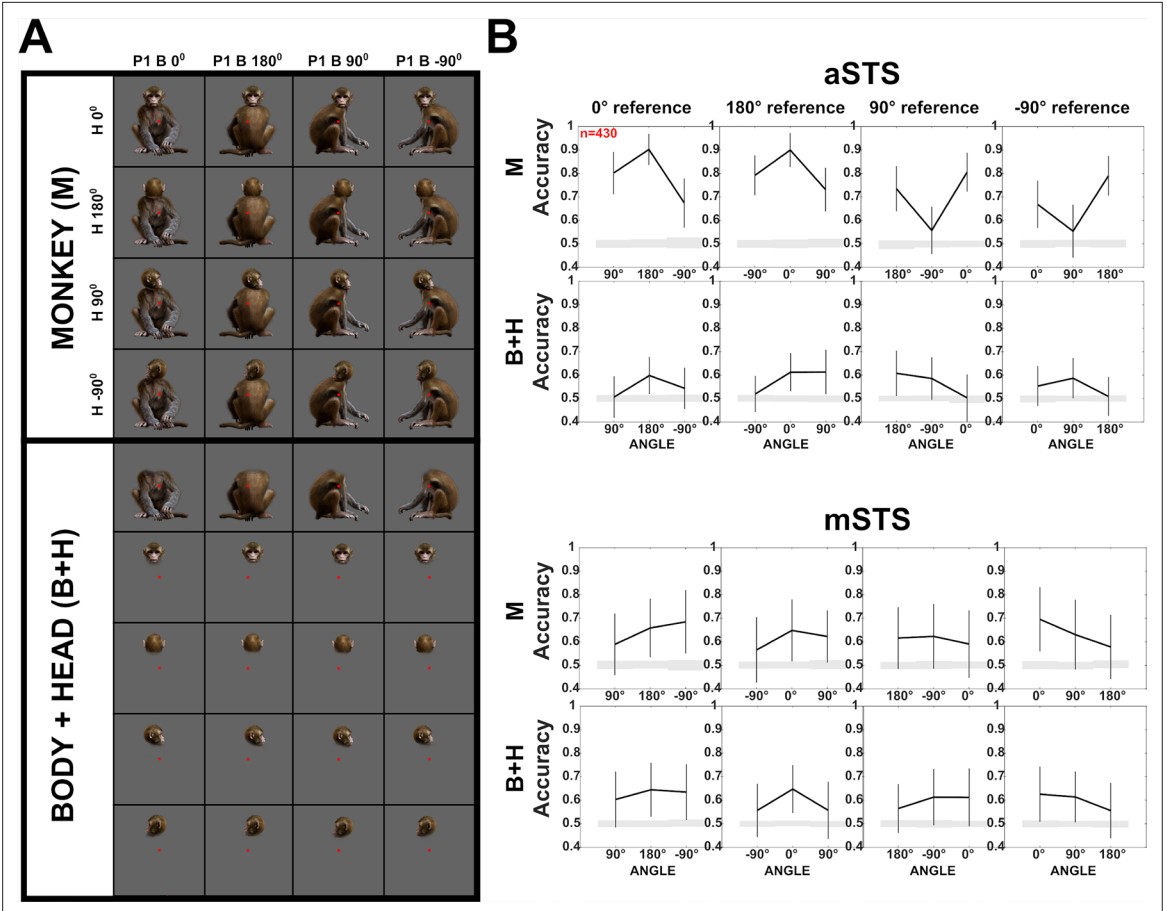

**Figure 6.** Decoding of head–body angles in the Sum versus Configuration test (experiment 2). (**A**) Stimuli of experiment 2: heads and bodies presented at four orientations (frontal, back, and the two lateral ones; B+H), along with all head–body configurations (M). Images were monkey-centered and the isolated heads and bodies were positioned to match their corresponding avatars. The red square indicates the fixation target. (**B**) Decoding accuracies for head–body configurations (M) and head–body sums (B+H) with zero, straight, leftward, and rightward right angles of head and body (trained pooling the four orientations (**A**)), for anterior superior temporal sulcus (aSTS) and mid superior temporal sulcus (mSTS), *n* indicates the number of units. Same conventions as *Figure 4*.

The online version of this article includes the following figure supplement(s) for figure 6:

**Figure supplement 1.** Decoding of head–body angles for head- and body-selective anterior superior temporal sulcus (aSTS) units (experiment 2).

presented in isolation. We compared the responses to 16 head–body orientation configurations to the responses to the heads and bodies shown in isolation with the same size and at the same visual field location as in the monkey configuration. We recorded with V-probes in mSTS and aSTS, overlapping the monkey patches defined by fMRI (*Figure 1A*; Materials and methods). The stimulus set included 16 MC images of the P1 avatar, crossing 4 head orientations and 4 body orientations (the frontal, the back view, and the two lateral views), yielding 16 conditions. In addition, we included images of the isolated heads, presented at the same position as in the avatar (16 conditions), and of the 4 body orientations (*Figure 6A*).

Spike sorting yielded 497 aSTS (M1 = 280; M2 = 217) and 1856 mSTS (M1 = 804; M2 = 1052) units that responded selectively to the 16 monkey conditions (Materials and methods). We decoded the head–body orientation angle for differences of 90° and 180° between angles of the head and body orientation (zero vs. right angle, zero vs. straight angle, etc.), using the same procedure as in experiment 1. We found a highly similar pattern (*n* = 430 units; equated across the two regions and subjects; randomly sampled per subject per resampling; *Figure 6B*) compared to that obtained with the smaller sample of single units of experiment 1 (*Figure 4B*). For aSTS, the classification accuracy for the zero versus straight angles was about 0.9 (*Figure 6B*). Again, classification accuracy for the right versus left angles, differing only in their sign, was much lower than that obtained for zero versus right angle, or

straight versus left angle, differing in absolute angle (*Figure 6B*; p < 0.01). The classification accuracy for mSTS units was again lower than for aSTS (*Figure 6B*), although no generalization across centering conditions was required for this decoding. Importantly, in mSTS there was no difference in decoding accuracy for the zero versus straight angle and right versus left angle. The reliability of the responses of the units was similar for aSTS (median Spearman–Brown corrected split-half Pearson correlation; M1: 0.70; M2: 0.69) and mSTS (median corrected *r*: M1: 0.72; M2: 0.71) and therefore cannot explain the difference in classification accuracy between the two regions.

Having replicated the main findings of experiment 1 (*Figure 4B*), we assessed whether the simultaneous presentation of the head and body was necessary to decode the angle between head and body orientation. To do this, we summed the net responses to the isolated head and body, presented at the same locations as in the whole-monkey image, for individual trials, thus obtaining 8 head–body sums per orientation and angle (zero, straight, and the left and right angles) of head and body orientation (*Figure 6A*). We decoded the angle using the face–body sums, employing the same procedure and units as for the whole monkey. For aSTS, the classification accuracies for the sums were markedly lower than those obtained for the monkey images (*Figure 6B*), and no difference in classification accuracy between the zero versus straight angles and left versus right angles was present for the sums (all p's > 0.4). The lower classification accuracy for the head–body sums is not due to a lower reliability of the summed responses compared to the whole-body responses (Spearman–Brown corrected split-half Pearson correlations; median whole-body *r*: M1: 0.7; M2: 0.69; sums *r*: M1: 0.76; M2: 0.66). These data suggest that the difference in classification accuracy between the zero versus straight angles and left versus right angles requires the simultaneous presence of the head and body. For mSTS, the classification accuracies for the head–body sums were similar to those obtained for the monkey images (*Figure 6B*), suggesting no genuine encoding of the head–body angle in this region.

## Head- and body-selective aSTS units show similar decoding patterns of head–body orientation angle (experiment 2)

To assess whether the head/body selectivity matters for the decoding of the head–body orientation angle, we subdivided our aSTS units into groups based on their head/body selectivity (Materials and methods; *Figure 6—figure supplement 1*). Head- and body-selective aSTS units showed, as a population, a similar decoding bias for zero versus straight angles and both confused the left and right angles (*Figure 6—figure supplement 1*).

## Effect of body inversion on responses and view selectivity (experiment 3)

We determined whether body inversion impairs the decoding accuracy of the head–body orientation angle. Inverting the body affects the global body configuration, keeping the local features intact (except for a 180° change). First, we assessed for the first time whether STS neurons show a body-inversion effect in response strength and/or capacity to discriminate between views of a pose. For this, we obtained a new dataset of aSTS and mSTS responses to the upright and inverted poses of the avatar. We employed the same two poses as for the single units (*Figure 7—figure supplement 1*). For each pose, we presented eight orientations (steps of 45°) of four head–body orientation angles (zero, straight, and left and right angles) and this upright and inverted (*Figure 7—figure supplement 1*). The images were MC, with the heads shifted toward the fixation point, as in experiment 1 (Materials and methods; *Figure 7—figure supplement 1* for examples). In addition, we included heads and bodies shown in isolation for a subset of these conditions (Materials and methods; *Figure 7—figure supplement 1*).

We recorded with V-probes at the same mSTS and aSTS locations (three anterior–posterior levels in each region) as in experiment 2 (*Figure 1A*). First, we selected for each pose those units that responded to the head, headless body, or monkey, either upright or inverted (Materials and methods), having the anatomically possible zero head–body angle (see *Figure 7—figure supplement 1* for examples). Then, for each pose, we selected the orientation (preferred orientation) that produced the maximum response for the stimuli corresponding to that orientation, irrespective of whether the stimuli were upright or inverted. The latter ensured an unbiased selection of the preferred orientation with respect to the upright versus inversion variable. We observed for each pose, in each region of each subject, a higher mean response for the upright compared to the inverted preferred orientation

of the whole monkey (Wilcoxon signed-rank tests on data pooled across monkeys: p's in *Figure 7C*; population PSTHs (Peri-Stimulus Time Histograms) in *Figure 7—figure supplement 2*). Selecting units that responded either to the upright or inverted heads showed less consistent inversion effects for the head than observed for the monkey: upright heads showed across subjects consistently and significantly greater responses than to inverted heads only for P2 in mSTS but not for the other pose or region (*Figure 7—figure supplement 3*). The same held for the headless body for those units that responded to either the upright or inverted bodies (*Figure 7—figure supplement 3*). Thus, the inversion effect was consistently present for the head–body configurations, and less so for the face and body only.

Behaviorally, the body-inversion effect corresponds to a higher discriminability of upright bodies. To assess this for our STS neurons, we decoded the eight whole-body orientations employing units responding to the whole-body images with a zero body–head angle, either upright or inverted. The decoding was performed separately for upright and inverted images. This was done for each pose, drawing randomly in each of the 1000 resamplings 60 units per region (Materials and methods). The mean decoding accuracy was significantly higher (all p-values <0.01; Materials and methods) for upright compared to inverted monkeys for both poses in aSTS (P1: upright versus inverted: 0.89 vs. 0.82; P2: 0.89 vs. 0.71, with standard deviations SD = 0.02) and mSTS (P1: 0.89 vs. 0.79; P2: 0.93 vs. 0.75; SDs ranging between 0.01 and 0.02).

## Decoding of head–body orientation angle is impaired for inverted bodies (experiment 3)

Next, we asked whether the decoding of the head–body orientation angle is affected by inversion. For each pose, we selected units that responded to our stimulus set of eight orientations of four angles for the upright or inverted avatar (*Figure 7A*; Materials and methods). Then, we decoded the head–body orientation angle from the aSTS units using the same classification procedure as above. For both poses ($n$ = 460 responsive aSTS units; an equal number of units randomly sampled per subject per resampling), the upright images showed again a higher classification performance for the zero versus straight angles and the lowest for the left versus right angles (*Figure 7B*). The classification accuracy dropped significantly (zero versus straight angle; all p-values <0.01; $n$ = 100 resamplings; Materials and methods) for the inverted bodies (*Figure 7B*), and this drop in classification accuracy tended to be larger for the zero versus straight angle decoding than for the left versus right angle decoding (the difference between upright and inverted left versus right angle decoding not significant for the two poses; p = 0.2 and 0.18). Thus, the decoding of the head–body orientation angle is impaired when inverting the monkey image.

The stronger inversion effect for the zero versus straight angles decoding suggests that the head–body angle signal in aSTS depends on the global head–body configuration and is not driven merely by inversion-invariant local features. This aligns with the marked generalization across head–body configuration orientation that we observed in experiment 1 for P1 (training using four orientations and testing using the other four); such across-orientation generalization would not be expected if local features of individual images were employed by the classifier.

## Discussion

We showed here that head and body orientations interact in ventral bank STS neurons, with some neurons being tuned to a particular combination of head and body orientation. Importantly, the head–body orientation angle, and not only the orientations of the head and the body per se, can be robustly decoded from a population of STS neurons. This decoding was greater for aSTS compared to mSTS neurons. There is a marked bias in the decoding of head–body orientation angle from aSTS with better decoding of absolute angle differences (e.g. 0° and 180° angles). Head–body orientation angles that differ only in their sign with respect to the anatomical plane (e.g. 90° vs. –90°angles) can be decoded less from aSTS neurons. Also, we provide here the first demonstration of an inversion effect for a body with a head in the macaque STS: the response strength, orientation decoding, and head–body orientation angle decoding are less for inverted compared to upright monkey avatars.

We could decode the orientation of the head–body configurations, which is an ecologically important property. From the same population of neurons, we could also decode the angle between

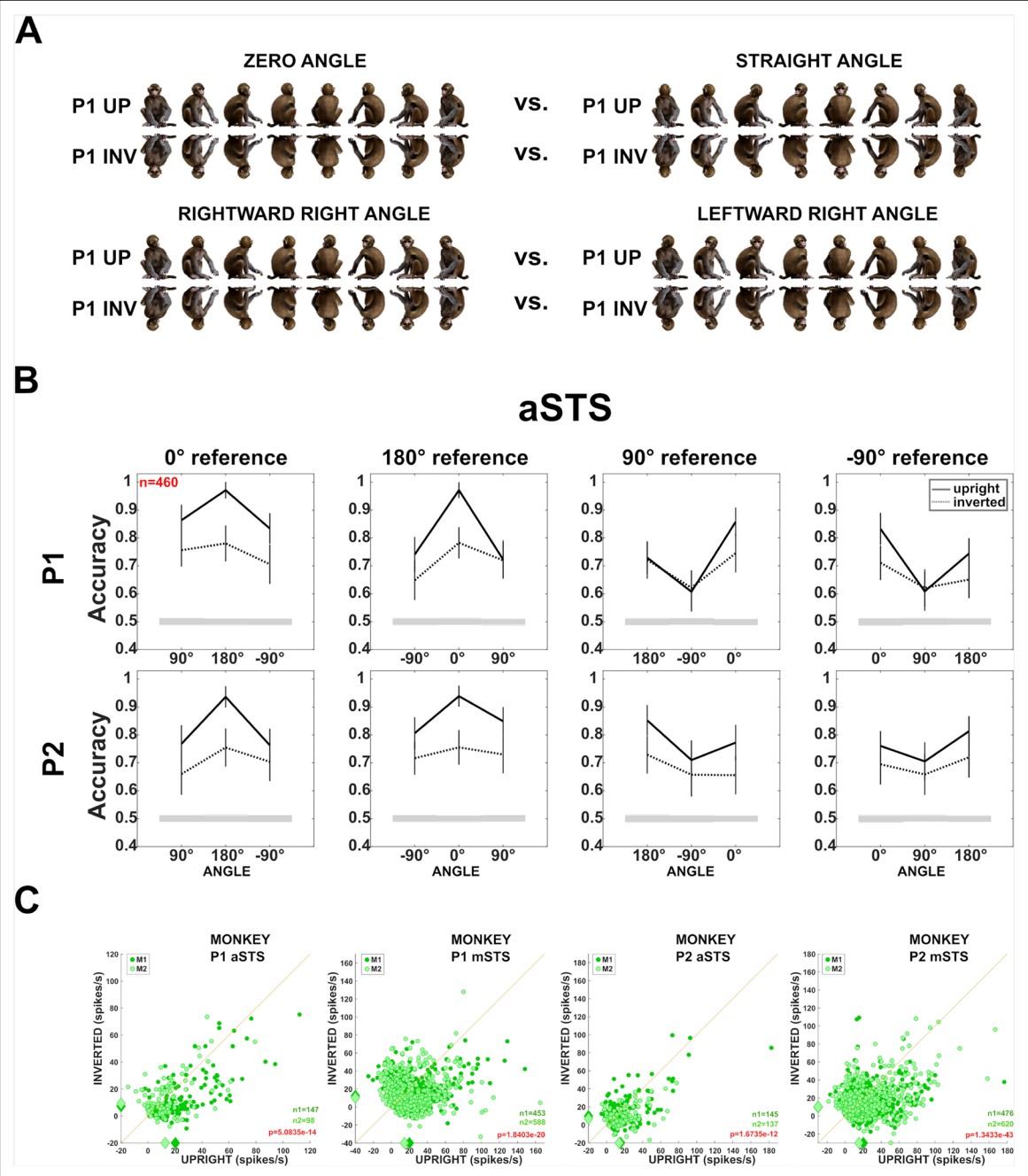

**Figure 7.** Effect of head–body configuration inversion (experiment 3). (**A**) Stimuli of the head–body configurations of P1 with zero and straight angles, and leftward and rightward right angles, at eight orientations, upright and inverted. (**B**) Decoding accuracies for the head–body configurations with zero, straight, leftward, and rightward right angles in anterior superior temporal sulcus (aSTS), for poses P1 (top panels) and P2 (bottom panels), *n* indicates the number of units. Solid and dotted lines represent data for upright and inverted configurations, respectively. Same conventions as *Figure 4*. (**C**) Responses to the preferred orientation of upright or inverted head–body configurations for aSTS and mid superior temporal sulcus (mSTS) and P1 and P2 (M1 (dark green); M2 (light green)), for the monkey-responsive units. Medians (diamonds) and the number of units per subject (M1: n1; M2: n2) are shown in corresponding colors. p-values of Wilcoxon signed-rank tests (data pooled across subjects) testing the difference in response between upright and inverted are shown (red: significant).

The online version of this article includes the following figure supplement(s) for figure 7:

**Figure supplement 1.** Monkey avatar stimuli for the Upright versus Inverted test (experiment 3).

**Figure supplement 2.** Population PSTHs of anterior superior temporal sulcus (aSTS) and mid superior temporal sulcus (mSTS) units recorded in the Upright versus Inverted test (experiment 3).

*Figure 7 continued on next page*

*Figure 7 continued*

**Figure supplement 3.** Inversion effect of head- and body-responsive anterior superior temporal sulcus (aSTS) and mid superior temporal sulcus (mSTS) units (experiment 3).

the head and the body, irrespective of the orientation of the configuration (*Figure 4A*). This can be viewed as a sensitivity for a particular head–body configuration that is defined by the orientation of the head *and* body, and is independent of the overall orientation of the agent. Remarkably, decoding of head–body configuration in the aSTS confused to a greater extent configurations in which the head and body orientation angles differ only in their sign (e.g. head turned to the left vs. to the right with respect to the body). The latter is quite a robust effect in the aSTS: we observed it in three different experiments. Images, especially of the standing pose, that differ only in the sign of the head–body angle are also to some extent mirror-symmetric. Our population of single aSTS neurons showed evidence of a lower sensitivity of mirror-symmetrical views for heads and bodies. Thus, one factor that may have contributed to the lower decoding of head–body orientation angles that differ only in sign is mirror symmetry. The highest decoding of the head–body angle occurred for the zero versus straight angles. Basically, this is a classification of a frequently observed head–body configuration, that is the same orientation of head and body, versus an anatomically impossible angle, at least for primates. In fact, all well-decodable absolute differences in angles consisted of a possible versus an impossible angle, while angles that differ only in their sign are either both anatomically possible or both impossible. Thus, this anatomical possibility factor may explain the better decoding for configurations that differ in their absolute angle. However, dissociating this factor from absolute versus signed angle difference, or mirror symmetry, is not trivial, since these three factors covary in monkey configurations when manipulating the orientation of the head and body.

The decoding of whole-monkey orientation, which can be based on head and/or body orientation, was present in mSTS and aSTS, which aligns with previous studies (*Bao et al., 2020*; *Freiwald and Tsao, 2010*; *Kumar et al., 2019*). In line with previous studies (*Kumar et al., 2019*; *Meyers et al., 2015*), the decoding of head and body orientation was better in mSTS compared to aSTS (*Figure 4— figure supplement 3*). However, the decoding of head–body configuration, that is head–body orientation angle, showed the opposite trend: it was worse in mSTS than in aSTS. This agrees with previous work that suggested the presence of head–body integration in the anterior but less so in the posterior temporal cortex (*Fisher and Freiwald, 2015*; *Hu et al., 2020*; *Zafirova et al., 2022*).

We observed no significant effect of head–body orientation angle on mean response strength, indicating that anatomically possible and impossible configurations elicited statistically similar averaged responses. This differs from a previous study in which responses to head–body configurations in which the body and head were misaligned were lower than those for a natural head–body configuration (*Zafirova et al., 2024*). However, that study targeted a patch that was defined in fMRI by contrasting aligned versus misaligned head–body configurations, which only partially overlapped the fMRI-defined monkey patch of the present study. Furthermore, the present study examined head–body orientation interactions, while the previous one head–body relative position interactions, which are fundamentally different. Nonetheless, the present study suggests that mSTS and aSTS neurons respond equally strongly to anatomically possible and impossible head–body configurations, at least when these are defined by the head–body orientation angle. However, as a population, they can distinguish strikingly well the possible and impossible configurations since these differ in their absolute angle (see above).

Our monkeys were performing a passive fixation task during the recordings and thus we do not have behavioral data on the discrimination of head–body orientation angles. Based on our electrophysiological data, one would predict that, at the behavioral level, the discrimination of head–body orientation angle, irrespective of the viewpoint of the avatar, would be more accurate for zero versus straight angles compared to right versus left angles, but this remains to be tested psychophysically.

Our work demonstrates for the first time an inversion effect for bodies *with* heads in the ventral bank of the STS. Previous studies reported a higher response to upright compared to inverted bodies in the STS (*Ashbridge et al., 2000*; *Popivanov et al., 2015*), but the neurons in these studies were searched with upright bodies. However, in our study, a search bias cannot explain the body-inversion effect since we selected responsive units using both upright and inverted images. A body-inversion effect has been observed in both humans and monkeys at the behavioral level (*Griffin and Oswald, 2022*; *Matsuno and Fujita, 2018*). Some but not all human studies (see *Griffin and Oswald, 2022* for

a meta-analysis) observed a stronger inversion effect for a body with than without a head. Our data suggest also a more consistent inversion effect for head–body configurations compared to headless bodies and heads in STS.

The body-inversion effect likely results from greater exposure to upright than inverted bodies during development, although, as for the face-inversion effect, this remains to be proved (*Kobylkov and Vallortigara, 2024*). Our monkeys were not exposed to the anatomically impossible head–body orientations, but still, the responses to these configurations were similar to those to anatomically possible configurations. This apparent discrepancy between the head-inversion effect and the anatomically (im)possible head–body orientation configuration may result from both head and body being in an infrequent orientation in the case of inversion while for the impossible head–body orientation configuration either the head or the body, but not both, might be at an infrequently exposed orientation.

In conclusion, we demonstrated that head and body orientations interact in the macaque STS. The population of aSTS neurons can provide a reliable signal for the recognition of particular head–body angles, especially between those that differ in their absolute angle with respect to the anatomical midline. This signal is less reliable for inverted head–body configurations, showing a strong inversion effect.

## Materials and methods
### Subjects
The experiments were performed with the two male rhesus monkeys (M1 and M2; *Macaca mulatta*, 6–7 years, 11–12 kg; Supplier: BPRC, Rijswijk, The Netherlands) which took part in our fMRI study (*Zafirova et al., 2022*). They were implanted with an MR-compatible plastic headpost for fixing the head during training, scanning, and recording, and a plastic custom-made recording chamber allowing a slightly oblique approach to the fMRI-defined monkey patches (for details, see Data acquisition). All experimental procedures and animal care complied with national and European regulations and were approved by the Animal Ethical Committee at KU Leuven (reference number: 105/2019).

### Data acquisition
Before the three electrophysiological experiments, reported here, we performed a functional MRI experiment on the same monkey subjects to determine the location of patches activated more by images of monkeys compared to objects (for details see *Zafirova et al., 2022*). The fMRI localizer had a block design with a Brown-noise backgrounds-only condition (baseline) and the following 6 conditions, 20 images each: monkeys, monkey faces, monkey headless bodies, and three groups of balanced manmade objects for the monkeys, the faces and the bodies, respectively (centered on the Brown-noise backgrounds). We employed the contrast monkeys versus monkey control objects (p < 0.05, Family-wise error corrected) to define the monkey-selective regions (monkey patches) in each subject's IT. In experiment 1, we targeted single units in the monkey patches in the ventral bank of the STS, lateral to the fundus, at two anterior–posterior levels (*Figure 1A*). In experiments 2 and 3, we recorded from a wider region, which overlapped with the two monkey patches and the recording locations of experiment 1. This was done in the context of another electrophysiological study in which we mapped the responses to heads and body parts. In these two experiments, the penetrations were at three anterior–posterior levels, spaced 1 mm apart in aSTS and mSTS. At each anterior–posterior level, we made penetrations in 1 mm steps from the lip of the STS toward the fundus (including the ventral part of the fundus). This resulted in a total of 18 and 27 penetrations in aSTS and mSTS, respectively, for both subjects. We recorded from the left hemisphere of M1 and the right hemisphere of M2 in all experiments.

Before the recording experiments, the subjects underwent an anatomical MRI (3.0T full-body Siemens scanner Magnetom Prisma Fit; magnetization-prepared rapid acquisition with gradient echo sequence; 0.6 mm isotropic voxel resolution). We inserted into the recording chamber grid (until the dura) long plastic capillaries filled with MRI opaque copper sulfate ($CuSO_4$) at predetermined positions and filled the recording chamber with a Gadoteric Acid (Dotarem) solution for visualization of the chamber and the grid. The activations were co-registered to the anatomical scans (Freesurfer http://surfer.nmr.mgh.harvard.edu/; FSLeyes, http://git.fmrib.ox.ac.uk/fsl/fsleyes/fsleyes/, SPM12, RRID:SCR_007037) and the co-registration was verified by visual examination. The recording locations

in the sagittal and coronal planes were verified with structural MRIs that visualized the guide tube tracks. The dorsal/ventral locations were based on observation of the white/gray matter transitions and silence associated with the STS during the recordings.

The position of one eye was continuously tracked with an infrared video-based tracking system (SR Research EyeLink; sampling rate 1 kHz). Stimuli were displayed on a VPixx LCD display (1920 × 1080 resolution; 120 Hz) at a distance of 57 cm from the subjects' eyes. The onset and offset of the stimulus were signaled utilizing a photodiode detecting luminance changes of a small square in the corner of the display (not visible to the animal), placed in the same frame as the stimulus events. A digital signal processing-based computer system developed in-house controlled stimulus presentation, event timing, and juice delivery while sampling the photodiode signal, vertical, and horizontal eye positions, spikes, and behavioral events.

Single-unit recordings in experiment 1 were performed with epoxylite-insulated tungsten microelectrodes (FHC) lowered with a Narishige microdrive into the brain using a stainless-steel guide tube that was fixed in a custom-made grid positioned within the recording chamber. After amplification and filtering, spikes of a single unit were isolated online using a custom amplitude- and time-based discriminator.

Spiking activity was recorded in experiments 2 and 3 with 24-channel V-probes (Plexon). The V-probe was lowered into the brain through a stainless-steel guide tube with a Narishige microdrive. The same grids were employed in experiment 1. After reaching the recording target, we waited for at least 30 min before the tests were run to obtain stable recordings. During that waiting period, the monkey could watch movies. We employed an INTAN RHD recording system to record spiking activity. The signal of each channel was bandpass-filtered online between 500 and 7500 Hz. The photodiode signal from the display was fed into the INTAN recording system, allowing alignment of stimulus events and the condition and trial data saved by the DSP-based computer system. We employed Plexon Offline Sorter to sort the spiking activity of each channel into single units and small clusters of (unsortable) units (multi-units). In this paper, we use the term 'units' for both single- and multi-units, combining single- and multi-units in the analyses of experiments 2 and 3. Sample size was determined based on previous studies examining STS units' selectivity for animate and inanimate objects.

## Stimuli and experiments

We report here three experiments designed to investigate the interaction between the orientations of the head and the body. The first experiment included a Selectivity test to assess the selectivity of the neurons recorded in the main Head–body Orientation test. The Selectivity test employed naturalistic grayscale static images centered on a uniform gray background. All subsequent tests employed static images of realistic 3D monkey avatar in color (https://www.turbosquid.com/3d-models/monkey-fur-rigged-1493219) on the same uniform gray background.

First, the avatar was positioned in two emotionally neutral natural monkey poses – a sitting and a standing one (P1 and P2, respectively; *Figure 2A*). Using a 3D avatar instead of actual monkey images allowed us to manipulate independently the orientation of the head and the body, for the two poses. The head and body orientations were obtained by rotation around the vertical axis of the avatar. The maximal vertical or horizontal extent of the images was set to 6°. Heads and headless bodies of some of the configurations were presented in isolation at their original location which allowed for direct comparison with the corresponding configurations. Details about the head and body orientations and the centering of the configurations for each experiment are described below.

## Experiment 1
### Selectivity test

To assess the category selectivity of the neurons of experiment 1, we employed 60 stimuli from a previous fMRI experiment (for the details about the stimuli, see *Zafirova et al., 2024*). In the Selectivity test, we presented 10 images of monkeys in diverse natural postures and orientations, together with 10 images of headless monkey bodies and 10 images of monkey faces, all cropped from different natural photographs of rhesus macaques. Additionally, we selected 10 control objects for each of the groups (examples in *Figure 1*). The objects matched the face-, body-, and monkey images as best as possible, as estimated by different shape-descriptive parameters (for details about the images and the matching procedure, see *Zafirova et al., 2024*). The low-level characteristics, like luminance and

contrast, were equated across the whole set (Matlab SHINE toolbox *Willenbockel et al., 2010*). The grayscale images were resized to fit a 6° × 6° stimulus window and gamma-corrected.

### Head–body orientation test

In this test, we manipulated the orientations of the head and the body independently, starting from the frontal view (0°) with eight 45° consecutive steps, generating 64 head–body combinations for the sitting and the standing pose (P1 and P2; *Figure 2A*). In terms of angle between the orientation of the head and the body, this resulted in eight head–body configurations with angles 0° (zero angle), 45°, 90° (rightward right angle), 135°, 180° (maximum possible absolute angle: straight angle), −135°, −90° (leftward right angle), and −45° between the head and the body. Each of these eight head–body configurations was spatially rotated with steps of 45° around its vertical axis, that is, each configuration was presented in eight different orientations. Ultimately, we had eight orientations of the head and the headless bodies (0° to −45° with steps of 45°), eight head–body configurations with angles 0° to −45° with steps of 45° angle between the head and the body, and eight different orientations of each of these configurations (rotated from 0° to −45° with steps of 45° around their vertical axis). As a convention, we will further refer to the orientation of the head–body configurations by the orientation of the body (torso). The head–body configurations with an angle between the head and the body more than an absolute angle of 90° were anatomically impossible.

The configurations were centered on the head (HC condition) or the monkey avatar (MC condition), for each of the two poses P1 and P2 (examples in *Figure 2B*). In the HC condition, the maximum extent of the bodies was 2.8° and 3.3° from the fixation point for P1 and P2, respectively. As we were interested in the possible head–body orientation interactions, and to ensure that the orientation of the head was clearly visible, the images were shifted 1° down in the lower visual field in the MC condition, so that the heads were closer to the fixation point. That way, the maximum extent of the head was 1.6° and 2.6° from the fixation point for P1 and P2, respectively, in the MC condition.

Additionally, heads and headless bodies from the configurations with zero and straight angles (0° and 180°; anatomically possible and impossible) between the head and the body of P1 and P2 were presented in isolation at the same locations as in the head–body configurations. We did that for four orientations of the two configurations: the frontal, the back, and the two lateral ones (0°, 180°, 90°, and −90°). We had four orientations of the headless bodies for the MC condition and four orientations of the head for the HC condition (one from each orientation; *Figure 2C*). These were compared to the head–body configuration with the same body orientation for the MC condition and to the head–body configuration with the same face orientation in HC conditions. We presented two head orientations – one from the zero angle and one from the straight angle configurations for each of the four body orientations in the MC condition, and two body orientations – one from the zero angle and one from the straight angle – for each of the four head orientations in the HC condition. Thus, we had four headless bodies and eight heads in the MC condition and four heads and eight bodies in the HC condition, for P1 and P2 (*Figure 2C*), or altogether standalone heads and bodies from the zero and straight angle configurations at four orientations, for each centering and pose. In total, we had 152 images ((64 + 12) for the MC and (64 + 12) for the HC condition) for each pose.

## Experiment 2

### Sum versus configuration test

In the second experiment, we presented only the sitting pose (P1) of the monkey avatar, centered on the whole body. Additionally, we employed four out of the eight head–body configurations: the zero angle (0°), the straight angle (180°), and the right and left angle configurations (90° and −90°). These were chosen because experiment 1 showed that the 0°–180° pair and the 90°–90° pair showed the largest difference in the decoding of the head–body orientation angle. Each of these configurations was presented at four orientations: the frontal, the back, and the two lateral ones (0°, 180°, 90°, and −90°), determined by the viewpoint of the body (rotated with steps of 90° around its vertical axis; *Figure 6A*).

As before, heads and headless bodies from the configurations were presented in isolation at their original locations. We did that for the four orientations. Thus, we had four orientations of the headless bodies. Here, we presented four head orientations from each of the four angles: the zero, the straight, and the left and right ones, resulting in overall 16 standalone heads and four headless bodies

(*Figure 6A*). Overall, the Sum versus Configuration test employed 36 images (16 configurations and 20 images of heads and headless bodies).

## Experiment 3
### Upright versus inverted test
In the last experiment, we presented both the sitting and the standing pose (P1 and P2) of the monkey avatar. We employed the same four head–body configurations as in experiment 2. Each of these configurations was presented at the eight orientations (body rotated with steps of 45° around its vertical axis; *Figure 5—figure supplement 1*). Crucially, we introduced one additional manipulation: all images were presented either upright or flipped around their horizontal axis (inverted; *Figure 7— figure supplement 1*). All were centered on the monkey avatar, but as in the Head–body Orientation test, we wanted to ensure that the orientation of the heads was visible. To that end, we shifted the upright images 1° down in the lower visual field and the inverted images 1° up in the upper visual field, so that the heads were closer to the fixation point (*Figure 5B*).

We presented also the heads and headless bodies from the zero and straight angle head–body configurations of the two poses in isolation at their original locations. We did that for four orientations: the frontal, the back, and the two lateral ones (0°, 180°, 90°, and –90°). Thus, we had four orientations of the headless bodies per pose. We presented two head orientations from each configuration head–body angle, resulting in four headless bodies and eight heads for a pose (*Figure 7—figure supplement 1*). Overall, this test included 176 images (the upright and inverted configurations of the two poses and their heads and bodies).

## Procedure
In experiment 1 (Selectivity and Head–body Orientation test), we recorded well-isolated single neurons, while single- and multi-units were recorded with V-probes in experiments 2 (Sum vs. Configuration test) and 3 (Upright vs. Inverted test). In each subject, we recorded in (experiments 1–3) and surrounding (experiments 2 and 3; see Results) patches that were activated more by images of monkeys compared to monkey control objects, in the anterior (aSTS) and mid ventral bank (aSTS) of the STS (*Figure 1A*). To control for potential order effects, we recorded first in aSTS in one subject, and the mSTS in the other for each experiment.

In experiment 1, we recorded in the Head–body Orientation test well-isolated single neurons that responded to monkeys, monkey faces *or* bodies in the preceding Selectivity test. All stimuli in all experiments were presented for 250 ms, with at least 600-ms interstimulus interval, on a uniform gray background. The subjects had to fixate a small (0.2° × 0.2°) red target. They received juice rewards for maintaining fixation in a 2° × 2° window around the target for 300 ms before the stimulus, during, and 300 ms after the stimulus. Trials in which the monkey failed to fixate continuously during this period were not analyzed further. The juice rewards were administered at fixed intervals in experiment 1 and after a completed trial in experiments 2 and 3. All stimuli of a test were presented pseudorandomly in blocks, and the order of the stimuli within a block was random. A stimulus of an aborted trial was presented again at a random moment during the same block.

## Data analyses
All analyses were performed only on those units for which we had 5 unaborted trials per stimulus for the Selectivity test and 8 unaborted trials per stimulus for all other experiments, and on net firing rates (FRs) unless stated otherwise. Net FRs were computed by subtracting the mean FR 100 ms before the stimulus onset (baseline window), from the FR in a response window of 50–300 ms after stimulus onset.

## Unit selection
### Experiment 1
To determine the responsiveness of a neuron, we computed mean FRs for each trial in two windows: a 200-ms baseline window that started 200 ms before stimulus onset and a 300-ms response window that started 50 ms after stimulus onset. We ran split-plot ANOVAs with within-trial factor baseline versus response window and between-trial factor stimulus. We considered as responsive only those neurons for which (1) either the main effect of the repeated factor or the interaction of the two factors

was significant (p < 0.05), and (2) their response was excitatory. Neurons that were irresponsive in the Selectivity test were excluded from the analyses of the Head–body Orientation test (4 out of 102 recorded aSTS and 5 out of 105 recorded mSTS neurons). The tests of the two poses were analyzed separately for responsiveness. Additionally, for each particular analysis, we excluded neurons that were irresponsive to all conditions of that analysis (e.g. separately for each pose and centering). These selections are described in the Results.

To compute the MSI (see below) we selected for each unit the monkey and the corresponding isolated head and body images (cases) for which there was a significant response to either the monkey, the head, or the body. The significance was tested with a Wilcoxon rank-sum test, comparing for each image the FR in baseline and response windows. The response needed to be excitatory to be considered for further analyses.

## Experiments 2 and 3

The responsive units of experiments 2 and 3 were selected using more stringent criteria since these were not isolated online using traditional single-unit recording methodology in which the electrode is moved for each unit to isolate it and responses are examined directly online. First, each unit was tested with a split-plot ANOVA with within-trial factor baseline versus response window and between-trial factor stimulus, as for experiment 1. Second, we employed a Kruskal–Wallis test (p < 0.05), performed on the mean raw FRs in the response window per trial and for all stimuli of a test, to exclude non-selective units. Third, only units with a z-score >3 for at least one stimulus were considered further. Z-scores for each stimulus were computed, as follows: first, we averaged the mean FRs in the baseline window (meanBaseStim) and the response window (meanRespStim) for the eight trials of each stimulus. Then, we calculated the mean (meanBaseAll) and the standard deviation of the baseline (stdBaseAll) across all stimuli. Finally, we subtracted the mean baseline from the averaged response of each stimulus and divided this difference by the standard deviation of the baseline: (meanRespStim − meanBaseAll)/stdBaseAll. The units in each analysis were subjected to the three criteria (split-plot ANOVA, the Kruskal–Wallis test, and the z-scoring).

## Reliability

We estimated the reliability of the responses of each responsive unit by randomly splitting for each stimulus the eight trials in two groups of four trials, and correlating the raw FR in the response window between the halves across stimuli. The random splitting was performed 100 times. The reliability corresponded to the mean of the 100 Spearman–Brown corrected Pearson correlation coefficients between the two halves. In some analyses, we normalized the correlation between two sets of responses (e.g. two centerings) by the reliability of the two responses by dividing the original correlation by the geometric mean of the reliability of the two response sets.

## Monkey-sum index

For each orientation, pose, and centering combination in which the neuron responded significantly either to the monkey (M), head (H), or body (B) (see Unit selection), we computed an MSI, using net FRs, as follows:

MSI = (Response M − (Response B + Response H))/(abs(Response M) + abs(Response B + Response H)).

## Interaction of head and body orientation tuning

For each responsive single unit of experiment 1, we assessed statistically the interaction between head and body orientation for each pose and centering with a two-way ANOVA with factors of head and body orientation. We performed two ANOVAs, one using the raw spike counts per trial and a second one using the following transformation of the raw spike counts per trial: square root (spike count + 3/8). The latter serves to stabilize the variance across conditions for Poisson-like distributions. Both produced similar results. We report the outcome of the ANOVAs using the square root transformed data. We isolated the interaction of head and body orientation for each neuron by computing the residuals of the square-root transformed responses from an additive model in which the responses to a head–body configuration are the sum of the mean responses to the body (computed by averaging responses across the eight head orientations) and head (averaged across the eight body orientations)

orientations. The additive model response to the monkey configuration with a head–body orientation (i, j) was modeled as:

Mean response head orientation (i) + mean response body orientation (j) – mean response to the 64 stimuli.

The residuals were computed for both centerings and then correlated across the centerings per neuron and pose combination.

## Decoding analyses

We performed a series of decoding analyses (see Results) on the raw responses unless stated otherwise with custom Matlab code employing a linear kernel SVM classifier. For the binary classification (pairs of head–body orientation angles) we employed the Matlab function *fitcsvm*. For multiclass classification, we employed the Matlab function *fitcecoc*. The regularization parameter C was 1 (default) in all cases. Per stimulus, we produced 8 pseudo-population vectors after permuting the order of the 8 trials per unit. We performed eightfold cross-validation, training on 7/8 of the pseudopopulation vectors, and the remaining ones were employed for testing. The reported classification scores are means of the percent correct classifications for the test vectors of 100 (or 1000; see below) resamplings of trials (experiments 1–3) and units (experiments 2 and 3). For the main decodings, we ran the same decoding procedure after permuting the stimulus labels, creating null distributions of chance performance. These were run 200 times, using the same number of resamplings. We plot the range of the mean classification scores (mean of 100 resamplings) obtained after label permutation in the figures.

### Decoding of head–body orientation angle

We decoded the head–body angle for pairs of angles (e.g. zero vs. the maximum possible straight angle), for all possible orientations of the respective head–body configuration. To do this, we created two classes of trials, each containing responses to the same angle (e.g. zero vs. straight angle). The head and body orientations per se were equal between the two classes. Since we had 8 trials of each of 8 (experiments 1 and 3) or 4 (experiment 2) head (and body) orientations per class, there were 64 trials per class in experiments 1 and 3, and 32 trials per class in experiment 2. For each orientation of a certain angle, we kept 7 trials for training, yielding in total 56 (7 × 8 orientations; experiments 1 and 3) or 28 training trials (experiment 2) per class. Test trials consisted of one trial per orientation, that is eight trials. This was repeated eight times for each resampling, each time using a different trial as a test trial, that is the training and testing trials were chosen pseudorandomly for each class separately. This procedure was critical to avoid unbalancing of head and body orientation between classes, which can produce artifactual results (e.g. classification scores well below chance levels). The pairs of angles employed in the decoding are described in the Results.

We will illustrate our approach for the binary classification of two head–body angles for experiment 1: zero versus straight angle (0° vs. 180°; *Figure 4A*). We pooled 8 trials of the 8 orientations of each angle (e.g. zero-angle configuration shown at a 0° (frontal) orientation, 45°, 90° (right lateral), 135°, 180°(back view), –135°, –90° (left lateral), and –45°; *Figure 4A*), providing 64 trials per angle. We employed an eightfold cross-validation approach, sampling in each cross-validation step 7 trials of each orientation per angle, ensuring balanced sampling across orientations. The same was done for the other angle (e.g. straight angle; *Figure 4A*). These 2 × 56 trials were then employed to train the SVM classifier. Testing was performed using one trial per orientation per angle (in total eight trials).

To assess the significance of the differences in classification scores between pairs of angles (e.g. decoding of zero vs. straight angle compared to decoding of the leftward vs. rightward right angles) we computed the difference in classification score between the two pairs for each resampling and the percentile of a difference of zero corresponded to the p-value. The significance of a two-tailed test requires this p-value to be smaller than 0.025. Note that the same set of neurons was taken per resampling when comparing differences in classification accuracy between conditions.

In experiment 1, we performed two types of decoding: (1) cross-centering decoding, where responses to MC images were used for training, and responses to HC images for testing (and vice versa), and (2) within-centering decoding, where both training and testing were within the same centering condition. We report the outcome of each. Additionally, we conducted decodings using responses to four orientations for training (e.g. frontal, back, and the two lateral views; '+' orientations)

and responses to the other four orientations for testing (e.g. the oblique views 45°, 135°, –135°, and –45°; 'x' orientations), and vice versa (*Figure 5A*).

For experiment 2, we performed also decoding of the summed responses to the isolated bodies and heads. Thus, for each head–body configuration, we summed the response to the corresponding head and body, using trials in which these stimuli were presented in isolation. Then, the decoding was run on the summed responses, using the same procedure as described above for the responses to monkey images, with two exceptions. First, as we employed summed head–body responses, in this decoding analysis we used net FRs (baseline subtracted) instead of raw responses, to avoid inflating the response when computing the sums. Second, to prevent unbalancing of head and body orientations across classes, testing trials were selected pseudorandomly to ensure that head and body orientations were equally represented in the training trials for both classes. The same balancing during cross-validation and the use of net FRs was also the case for the decoding of the angles for the monkey configurations (simultaneous presence of head and body) in experiment 2.

## Decoding of orientation

The main decodings in experiment 3 were conducted in line with the described default procedure. Additionally, in experiment 3, we decoded the orientation of the monkey for the zero-angle head–body configuration. This was done separately for the upright and inverted poses. We employed eight-fold cross-validation, using 7 trials per orientation for training and one per orientation for testing. In each of the 1000 resamplings, we randomly sampled 60 units from the population of units (aSTS P1 $n$ = 332, P2 $n$ = 369; mSTS P1 $n$ = 1298, P2 $n$ = 1332; equated per monkey). This number of units (equal to that of the experiment 1 decoding analyses) was used to avoid ceiling effects. The mean of 10 classification accuracies (i.e. of 10 resamplings) was employed to obtain a distribution ($n$ = 100) of the differences in classification accuracy between upright and inverted images. The zero-difference value fell outside the distribution of each of the two regions and poses. The reported standard deviations of the classification accuracies are computed using also the means of 10 resamplings.

## Acknowledgements

The authors thank I Puttemans, A Hermans, W Depuydt, C Ulens, S Verstraeten, J Helin, ST Riyahi, M De Paep, and Y Gürsoy for technical and administrative support. This research was supported by Fonds Wetenschappelijk Onderzoek (FWO) Vlaanderen (G0E0220N), KU Leuven grant C14/21/111, and the European Research Council (ERC) under the European Union's Horizon 2020 research and innovation program (grant agreement 856495).

## Additional information

### Funding

| Funder | Grant reference number | Author |
|---|---|---|
| Fonds Wetenschappelijk Onderzoek | G0E0220N | Rufin Vogels |
| European Research Council | 10.3030/856495 | Rufin Vogels |
| KU Leuven | C14/21/111 | Rufin Vogels |

The funders had no role in study design, data collection, and interpretation, or the decision to submit the work for publication.

### Author contributions

Yordanka Zafirova, Conceptualization, Data curation, Software, Formal analysis, Investigation, Visualization, Methodology, Writing – original draft, Writing – review and editing; Rufin Vogels, Conceptualization, Resources, Formal analysis, Supervision, Funding acquisition, Investigation, Methodology, Writing – original draft, Project administration, Writing – review and editing

## Author ORCIDs
Yordanka Zafirova (iD) http://orcid.org/0000-0002-2153-6926
Rufin Vogels (iD) https://orcid.org/0000-0002-8778-835X

## Ethics
All experimental procedures and animal care complied with national and European regulations and were approved by the Animal Ethical Committee at KU Leuven (Permit number 105/2019).

Reviewer #1 (Public review): https://doi.org/10.7554/eLife.105714.3.sa1
Reviewer #2 (Public review): https://doi.org/10.7554/eLife.105714.3.sa2
Reviewer #3 (Public review): https://doi.org/10.7554/eLife.105714.3.sa3
Author response https://doi.org/10.7554/eLife.105714.3.sa4

## Additional files

### Supplementary files
MDAR checklist

### Data availability
The data and code to generate the figures are available: Data: https://osf.io/7t2e8/. Codes: https://github.com/Yozafirova/head-body-orientation-interaction (copy archived at *Zafirova, 2024*).

The following dataset was generated:

| Author(s) | Year | Dataset title | Dataset URL | Database and Identifier |
| --- | --- | --- | --- | --- |
| Zafirova Y | 2024 | Head-body orientation interaction | https://osf.io/7t2e8/ | Open Science Framework, 7t2e8 |

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
