## [Editor Report · eLife Assessment]

This **important** study examines the neuronal mechanisms underlying visual perception of integrated face and body cues. The innovative paradigm, which employs monkey avatars in combination with electrophysiological recordings from fMRI-defined brain areas, provides **compelling** evidence on face and body integration. These results should be of wide interest to system and cognitive neuroscientists, psychologists, and behavioural biologists working on visual and social cognition.

---

## [Referee Report · Reviewer #1 (Public review)]

The study addresses how faces and bodies are integrated in two STS face areas revealed by fMRI in the primate brain. It is building upon recordings and analysis of the responses of large populations of neurons to three sets of images, that vary face and body positions. These sets allowed the author to thoroughly investigate invariance to position on the screen (MC HC), to pose (P1 P2), to rotation (0 45 90 135 180 225 270 315), to inversion, to possible and impossible postures (all vs straight), to presentation of head and body together or in isolation. By analyzing neuronal responses, they find that different neurons showed preferences for body orientation, or head orientation or for the interaction between the two. By using a linear support vector machine classifier, they show that the neuronal population can decode head-body angle presented across orientations, in the anterior aSTS patch (but not middle mSTS patch), except for mirror orientation. On the contrary, mSTS neurons show less invariance for head-body angle and more specialization for head or body orientation.

Strengths:

These results expand prior work on the role of Anterior STS fundus face area in face-body integration and its invariance to mirror symmetry, with a rigorous set of stimuli revealing the workings of these neuronal populations in processing individuals as a whole, in an important series of carefully designed conditions.

It also raises questions for future investigations comparing humans and monkeys expertise with upright and inverted configurations, in light of monkey-specific hanging upside-down behavior. Further, using two types of body postures (sitting, standing), they show a correlation in head-body angle between postures, suggesting that monkey orientation might be more meaningful to these neurons than precise posture.

---

## [Referee Report · Reviewer #2 (Public review)]

Summary:

This paper investigates the neuronal encoding of the relationship between head and body orientations in the brain. Specifically, the authors focus on the angular relationship between the head and body by employing virtual avatars. Neuronal responses were recorded electrophysiologically from two fMRI-defined areas in the superior temporal sulcus and analyzed using decoding methods. They found that: (1) anterior STS neurons encode head-body angle configurations; (2) these neurons distinguish aligned and opposite head-body configurations effectively, whereas mirror-symmetric configurations are more difficult to differentiate; and (3) an upside-down inversion diminishes the encoding of head-body angles. These findings advance our understanding of how visual perception of individuals is mediated, providing a fundamental clue as to how the primate brain processes the relationship between head and body-a process that is crucial for social communication.

Strengths:

The paper is clearly written, and the experimental design is thoughtfully constructed and detailed. The use of electrophysiological recordings from fMRI-defined areas elucidated the mechanism of head-body angle encoding at the level of local neuronal populations. Multiple experiments, control conditions, and detailed analyses thoroughly examined various factors that could affect the decoding results. The decoding methods effectively and consistently revealed the encoding of head-body angles in the anterior STS neurons. Consequently, this study offers valuable insights into the neuronal mechanisms underlying our capacity to integrate head and body cues for social cognition-a topic that is likely to captivate readers in this field.

Weaknesses:

I did not identify any major weaknesses in this paper.

---

## [Referee Report · Reviewer #3 (Public review)]

Summary:

Zafirova et al. investigated the interaction of head and body orientation in the macaque superior temporal sulcus (STS). Combining fMRI and electrophysiology, they recorded responses of visual neurons to a monkey avatar with varying head and body orientations. They found that STS neurons integrate head and body information in a nonlinear way, showing selectivity for specific combinations of head-body orientations. Head-body configuration angles can be reliably decoded, particularly for neurons in the anterior STS, suggesting a transformation of face/body orientation signals from the middle to the anterior STS. Furthermore, body inversion resulted in reduced decoding of head-body configuration angles. Compared to previous work that examined face or body alone, this study demonstrates how head and body information are integrated to compute a socially meaningful signal.

Strengths:

This work presents an elegant design of visual stimuli, with a monkey avatar of varying head and body orientations, making the analysis and interpretation straightforward. Together with several control experiments, the authors systematically investigated different aspects of head-body integration in the macaque STS. The results and analyses of the paper are convincing.

Weakness:

While this work has characterized the neural integration of head and body information in detail, it's unclear how the neural representation relates to the animal's perception. Behavioural experiments using the same set of stimuli could help address this question, but I agree that these additional experiments may be beyond the scope of the current paper.

---

## [Author Response]

The following is the authors’ response to the original reviews

**Public Reviews:**

**Reviewer #1 (Public review):**
Summary:The study addresses how faces and bodies are integrated in two STS face areas revealed by fMRI in the primate brain. It builds upon recordings and analysis of the responses of large populations of neurons to three sets of images, that vary face and body positions. These sets allowed the authors to thoroughly investigate invariance to position on the screen (MC HC), to pose (P1 P2), to rotation (0 45 90 135 180 225 270 315), to inversion, to possible and impossible postures (all vs straight), to the presentation of head and body together or in isolation. By analyzing neuronal responses, they found that different neurons showed preferences for body orientation, head orientation, or the interaction between the two. By using a linear support vector machine classifier, they show that the neuronal population can decode head-body angle presented across orientations, in the anterior aSTS patch (but not middle mSTS patch), except for mirror orientation.Strengths:These results extend prior work on the role of Anterior STS fundus face area in face-body integration and its invariance to mirror symmetry, with a rigorous set of stimuli revealing the workings of these neuronal populations in processing individuals as a whole, in an important series of carefully designed conditions.Minor issues and questions that could be addressed by the authors:(1) Methods. While monkeys certainly infer/recognize that individual pictures refer to the same pose with varying orientations based on prior studies (Wang et al.), I am wondering whether in this study monkeys saw a full rotation of each of the monkey poses as a video before seeing the individual pictures of the different orientations, during recordings.

The monkeys had not been exposed to videos of a rotating monkey pose before the recordings. However, they were reared and housed with other monkeys, providing them with ample experience of monkey poses from different viewpoints.

(2) Experiment 1. The authors mention that neurons are preselected as face-selective, body-selective, or both-selective. Do the Monkey Sum Index and ANOVA main effects change per Neuron type?

We have performed a new analysis to assess whether the Monkey Sum Index is related to the response strength for the face versus the body as measured in the Selectivity Test of Experiment 1. To do this we selected face- and body-category selective neurons, as well as neurons responding selectively to both faces and bodies. First, we selected those neurons that responded significantly to either faces, bodies, or the two control object categories, using a split-plot ANOVA for these 40 stimuli. From those neurons, we selected face-selective ones having at least a twofold larger mean net response to faces compared to bodies (faces > 2 * bodies) and the control objects for faces (faces > 2* objects). Similarly, a body-selective neuron was defined by a twofold larger mean net response to bodies compared to faces and the control objects for bodies. A body-and-face selective neuron was defined as having a twofold larger net response to the faces compared to their control objects, and to bodies compared to their control objects, with the ratio between mean response to bodies and faces being less than twofold. Then, we compared the distribution of the Monkey Sum Index (MSI) for each region (aSTS; mSTS), pose (P1, P2), and centering (head- (HC) or monkey-centered (MC)) condition. Too few body-and-face selective neurons were present in each combination of region, pose, and centering (a maximum of 7) to allow a comparison of their MSI distribution with the other neuron types. The Figure below shows the distribution of the MSI for the different orientation-neuron combinations for the body- and face-selective neurons (same format as in Figure 3a, main text). The number of body-selective neurons, according to the employed criteria, varied from 21 to 29, whereas the number of face-selective neurons ranged from 14 to 24 (pooled across monkeys). The data of the two subjects are shown in a different color and the number of cases for each subject is indicated (n1: number of cases for M1; n2: number of cases for M2). The arrows indicate the medians for the data pooled across the monkey subjects. For the MC condition, the MSI tended to be more negative (i.e. relatively less response to the monkey compared to the sum of the body and face responses) for the face compared to the body cells, but this was significant only for mSTS and P1 (p = 0.043; Wilcoxon rank sum test; tested after averaging the indices per neuron to avoid dependence of indices within a neuron). No consistent, nor significant tendencies were observed for the HC stimuli. This absence of a consistent relationship between MSI and face- versus body-selectivity is in line with the absence of a correlation between the MSI and face- versus body-selectivity using natural images of monkeys in a previous study (Zafirova Y, Bognár A, Vogels R. Configuration-sensitive face-body interactions in primate visual cortex. Prog Neurobiol. 2024 Jan;232:102545).

We did not perform a similar analysis for the main effects of the two-way ANOVA because the very large majority of neurons showed a significant effect of body orientation and thus no meaningful difference between the two neuron types can be expected.

**Author response image 1. sa4fig1:** 

(3) I might have missed this information, but the correlation between P1 and P2 seems to not be tested although they carry similar behavioral relevance in terms of where attention is allocated and where the body is facing for each given head-body orientation.

Indeed, we did not compute this correlation between the responses to the sitting (P1) and standing (P2) pose avatar images. However, as pointed out by the reviewer, one might expect such correlations because of the same head orientations and body-facing directions. Thus, we computed the correlation between the 64 head-body orientation conditions of P1 and P2 for those neurons that were tested with both poses and showed a response for both poses (Split-plot ANOVA). This was performed for the Head-Centered and Monkey-Centered tests of Experiment 1 for each monkey and region. Note that not all neurons were tested with both poses (because of failure to maintain isolation of the single unit in both tests or the monkey stopped working) and not all neurons that were recorded in both tests showed a significant response for both poses, which is not unexpected since these neurons can be pose selective. The distribution of the Pearson correlation coefficients of the neurons with a significant response in both tests is shown in Figure S1. The median correlation coefficient was significantly larger than zero for each region, monkey, and centering condition (outcome of Wilcoxon tests, testing whether the median was different from zero (p1 = p-value for M1; p2: p-value for M2) in Figure), indicating that the effect of head and/or body orientation generalizes across pose. We have noted this now in the Results (page 12) and added the Figure (New Figure S1) in the Suppl. Material.

(4) Is the invariance for position HC-MC larger in aSTS neurons compared to mSTS neurons, as could be expected from their larger receptive fields?

Yes, the position tolerance of the interaction of body and head orientation was significantly larger for aSTS compared to mSTS neurons, as we described on pages 11 and 12 of the Results. This is in line with larger receptive fields in aSTS than in mSTS. However, we did not plot receptive fields in the present study.

(5) L492 "The body-inversion effect likely results from greater exposure to upright than inverted bodies during development". Monkeys display more hanging upside-down behavior than humans, however, does the head appear more tilted in these natural configurations?

Indeed, infant monkeys do spend some time hanging upside down from their mother's belly. While we lack quantitative data on this behavior, casual observations suggest that even young monkeys spend more time upright. The tilt of the head while hanging upside down can vary, just as it does in standing or sitting monkeys (as when they search for food or orient to other individuals). To our knowledge, no quantitative data exist on the frequency of head tilts in upright versus upside-down monkeys. Therefore, we refrain from further speculation on this interesting point, which warrants more attention.

(6) Methods in Experiment 1. SVM. How many neurons are sufficient to decode the orientation?

The number of neurons that are needed to decode the head-body orientation angle depends on which neurons are included, as we show in a novel analysis of the data of Experiment 1. We employed a neuron-dropping analysis, similar to Chiang et al. (Chiang FK, Wallis JD, Rich EL. Cognitive strategies shift information from single neurons to populations in prefrontal cortex. Neuron. 2022 Feb 16;110(4):709-721) to assess the positive (or negative) contribution of each neuron to the decoding performance. We performed cross-validated linear SVM decoding N times, each time leaving out a different neuron (using N-1 neurons; 2000 resamplings of pseudo-population vectors). We then ranked decoding accuracies from highest to lowest, identifying the ‘worst’ (rank 1) to ‘best’ (rank N) neurons. Next, we conducted N decodings, incrementally increasing the number of included neurons from 1 to N, starting with the worst-ranked neuron (rank 1) and sequentially adding the next (rank 2, rank 3, etc.). This analysis focused on zero versus straight angle decoding in the aSTS, as it yielded the highest accuracy. We applied it when training on MC and testing on HC for each pose. Plotting accuracy as a function of the number of included neurons suggested that less than half contributed positively to decoding. We show also the ten “best” neurons for each centering condition and pose. These have a variety of tuning patterns for head and body orientation suggesting that the decoding of head-body orientation angle depends on a population code. Notably, the best-ranked (rank N) neuron alone achieved above-chance accuracy. We have added this interesting and novel result to the Results (page 16) and Suppl. Material (new Figure S3).

(7) Figure 3D 3E. Could the authors please indicate for each of these neurons whether they show a main effect of face, body, or interaction, as well as their median corrected correlation to get a flavor of these numbers for these examples?

We have indicated these now in Figure 3.

(8) Methods and Figure 1A. It could be informative to precise whether the recordings are carried in the lateral part of the STS or in the fundus of the STS both for aSTS and mSTS for comparison to other studies that are using these distinctions (AF, AL, MF, ML).

In experiment 1, the recording locations were not as medial as the fundus. For experiments 2 and 3, the ventral part of the fundus was included, as described in the Methods. We have added this to the Methods now (page 31).

Wang, G., Obama, S., Yamashita, W. et al. Prior experience of rotation is not required for recognizing objects seen from different angles. Nat Neurosci 8, 1768-1775 (2005). https://doi-org.insb.bib.cnrs.fr/10.1038/nn1600

**Reviewer #2 (Public review):**
Summary:This paper investigates the neuronal encoding of the relationship between head and body orientations in the brain. Specifically, the authors focus on the angular relationship between the head and body by employing virtual avatars. Neuronal responses were recorded electrophysiologically from two fMRI-defined areas in the superior temporal sulcus and analyzed using decoding methods. They found that: (1) anterior STS neurons encode head-body angle configurations; (2) these neurons distinguish aligned and opposite head-body configurations effectively, whereas mirror-symmetric configurations are more difficult to differentiate; and (3) an upside-down inversion diminishes the encoding of head-body angles. These findings advance our understanding of how visual perception of individuals is mediated, providing a fundamental clue as to how the primate brain processes the relationship between head and body - a process that is crucial for social communication.Strengths:The paper is clearly written, and the experimental design is thoughtfully constructed and detailed. The use of electrophysiological recordings from fMRI-defined areas elucidated the mechanism of head-body angle encoding at the level of local neuronal populations. Multiple experiments, control conditions, and detailed analyses thoroughly examined various factors that could affect the decoding results. The decoding methods effectively and consistently revealed the encoding of head-body angles in the anterior STS neurons. Consequently, this study offers valuable insights into the neuronal mechanisms underlying our capacity to integrate head and body cues for social cognition-a topic that is likely to captivate readers in this field.Weaknesses:I did not identify any major weaknesses in this paper; I only have a few minor comments and suggestions to enhance clarity and further strengthen the manuscript, as detailed in the Private Recommendations section.
**Reviewer #3 (Public review):**
Summary:Zafirova et al. investigated the interaction of head and body orientation in the macaque superior temporal sulcus (STS). Combining fMRI and electrophysiology, they recorded responses of visual neurons to a monkey avatar with varying head and body orientations. They found that STS neurons integrate head and body information in a nonlinear way, showing selectivity for specific combinations of head-body orientations. Head-body configuration angles can be reliably decoded, particularly for neurons in the anterior STS. Furthermore, body inversion resulted in reduced decoding of head-body configuration angles. Compared to previous work that examined face or body alone, this study demonstrates how head and body information are integrated to compute a socially meaningful signal.Strengths:This work presents an elegant design of visual stimuli, with a monkey avatar of varying head and body orientations, making the analysis and interpretation straightforward. Together with several control experiments, the authors systematically investigated different aspects of head-body integration in the macaque STS. The results and analyses of the paper are mostly convincing.Weaknesses:(1) Using ANOVA, the authors demonstrate the existence of nonlinear interactions between head and body orientations. While this is a conventional way of identifying nonlinear interactions, it does not specify the exact type of the interaction. Although the computation of the head-body configuration angle requires some nonlinearity, it's unclear whether these interactions actually contribute. Figure 3 shows some example neurons, but a more detailed analysis is needed to reveal the diversity of the interactions. One suggestion would be to examine the relationship between the presence of an interaction and the neural encoding of the configuration angle.

This is an excellent suggestion. To do this, one needs to identify the neurons that contribute to the decoding of head-body orientation angles. For that, we employed a neuron-dropping analysis, similar to Chiang et al. (Chiang FK, Wallis JD, Rich EL. Cognitive strategies shift information from single neurons to populations in prefrontal cortex. Neuron. 2022 Feb 16;110(4):709-721.) to assess the positive (or negative) contribution of each neuron to the decoding performance. We performed cross-validated linear SVM decoding N times, each time leaving out a different neuron (using N-1 neurons; 2000 resamplings of pseudo-population vectors). We then ranked decoding accuracies from highest to lowest, identifying the ‘worst’ (rank 1) to ‘best’ (rank N) neurons. Next, we conducted N decodings, incrementally increasing the number of included neurons from 1 to N, starting with the worst-ranked neuron (rank 1) and sequentially adding the next (rank 2, rank 3, etc.). This analysis focused on zero versus straight angle decoding in the aSTS, as it yielded the highest accuracy. We applied it when training on MC and testing on HC for each pose. Plotting accuracy as a function of the number of included neurons suggested that less than half contributed positively to decoding (see Figure S3). We examined the tuning for head and body orientation of the 10 “best” neurons (Figure S3). For half or more of those the two-way ANOVA showed a significant interaction. These are indicated by the red color in the Figure. They showed a variety of tuning patterns for head and body orientation, suggesting that the decoding of the head-body orientation angle results from a combination of neurons with different tuning profiles. Based on a suggestion from reviewer 2, we performed for each neuron of experiment 1 a one-way ANOVA with as factor head-body orientation angle. To do that, we combined all 64 trials that had the same head-body orientation angle. The percentage of neurons (required to be responsive in the tested condition) for which this one-way ANOVA was significant was low but larger than the expected 5% (Type 1 error), with a median of 16.5% (range: 3 to 23%) in aSTS and 8% for mSTS (range: 0-19%). However, a higher percentage of the 10 best neurons for each pose (indicated by the star) showed a significant one-way ANOVA for angle (for P1, MC: 50% (95% confidence interval (CI): 19% – 81%); P1, HC: 70% (CI: 35% - 93%); P2, MC: 70% (CI: 35% – 93%); P2: HC: 50% (CI: 19%-81%)). These percentages were significantly higher than expected for a random sample from the population of neurons for each pose-centering combination (expected percentages listed in the same order as above: 16%, 13%, 16%, and 10%; all outside CI). Thus, for at least half of the “best” neurons, the response differed significantly among the head-orientation angles at the single neuron level. Nonetheless, the tuning profiles were diverse, suggesting a populationl code for head-body orientation angle. We have added this interesting and novel result to the Results (page 16) and Suppl. Material (Figure S3).

(2) Figure 4 of the paper shows a better decoding of the configuration angle in the anterior STS than in the middle STS. This is an interesting result, suggesting a transformation in the neural representation between these two areas. However, some control analyses are needed to further elucidate the nature of this transformation. For example, what about the decoding of head and body orientations - dose absolute orientation information decrease along the hierarchy, accompanying the increase in configuration information?

We have performed now two additional analyses, one in which we decoded the orientation of the head and another one in which we decoded the orientation of the body. We employed the responses to the avatar of experiment 1, using the same sample of neurons of which we decoded the head-body orientation angle. To decode the head orientation, the trials with identical head orientation, irrespective of their body orientation, were given the same label. For this, we employed only responses in the head-centered condition. To decode the body orientation, the trials with identical body orientation, irrespective of their head orientation, had the same label, and we employed only responses in the body-centered condition. The decoding was performed separately for each pose (P1 and P2) and region. We decoded either the responses of 20 neurons (10 randomly sampled from each monkey for each of the 1000 resamplings), 40 neurons (20 randomly sampled per monkey), or 60 neurons (30 neurons per monkey) since the sample of 60 neurons yielded close to ceiling performance for the body orientation decoding. For each pose, the body orientation decoding was worse for aSTS than for mSTS, although this difference reached significance only for P1 and for the 40 neurons sample of P2 (p < 0.025; two-tailed test; same procedure as employed for testing the significance of the decoding of whole-body orientation for upright versus inverted avatars (Experiment 3)). Face orientation decoding was significantly worse for aSTS compared to mSTS. These results are in line with the previously reported decreased decoding of face orientation in the anterior compared to mid-STS face patches (Meyers EM, Borzello M, Freiwald WA, Tsao D. Intelligent information loss: the coding of facial identity, head pose, and non-face information in the macaque face patch system. J Neurosci. 2015 May 6;35(18):7069-81), and decreased decoding of body orientation in anterior compared to mid-STS body patches (Kumar S, Popivanov ID, Vogels R. Transformation of Visual Representations Across Ventral Stream Body-selective Patches. Cereb Cortex. 2019 Jan 1;29(1):215-229). As mentioned by the reviewer, this contrasts with the decoding of the head-body orientation angle, which increases when moving more anteriorly. We mention this finding now in the Discussion (page 27) and present the new Figure S10 in the Suppl. Material.

(3) While this work has characterized the neural integration of head and body information in detail, it's unclear how the neural representation relates to the animal's perception. Behavioural experiments using the same set of stimuli could help address this question, but I agree that these additional experiments may be beyond the scope of the current paper. I think the authors should at least discuss the potential outcomes of such experiments, which can be tested in future studies.

Unfortunately, we do not have behavioral data. One prediction would be that the discrimination of head-body orientation angle, irrespective of the viewpoint of the avatar, would be more accurate for zero versus straight angles compared to the right versus left angles. We have added this to the Discussion (page 28).

**Recommendations for the authors:**

**Reviewer #1 (Recommendations for the authors):**
(1) P22 L373. It should read Figure S5C instead of S4C.

Thanks; corrected.

(2) Figure 7B. All inverted decoding accuracies, although significantly lower than upright decoding accuracies, appear significantly above baseline. Should the title be amended accordingly?

Thanks for pointing this out. To avoid future misunderstanding we have changed the title to:

“Integration of head and body orientations in the macaque superior temporal sulcus is stronger for upright bodies”

(3) Discussion L432-33. "with some neurons being tuned to a particular orientation of both the head and the body". Wouldn't that be visible as a diagonal profile on the normalized net responses in Fig 3D? Or can the Anova evidence such a tuning?

We meant to say that some neurons were tuned to a particular combination of head and body orientation, like the third aSTS example neuron shown in Figure 3D. We have corrected the sentence.

**Reviewer #2 (Recommendations for the authors):**
Major comment:This paper effectively demonstrates that the angular relationship between the head and body can be decoded from population responses in the anterior STS. In other words, these neurons encode information about the head-body angle. However, how exactly do these neurons encode this information? Given that the study employed electrophysiological recordings from a local population of neurons, it might be possible to provide additional data on the response patterns of individual neurons to shed light on the underlying encoding mechanisms.Although the paper already presents example response patterns (Figures 3D, E) and shows that STS neurons encode interactions between head and body orientations (Figure 3B), it remains unclear whether the angle difference between the head and body has a systematic effect on neuronal responses. For instance, a description of whether some neurons preferentially encode specific head-body angle differences (e.g., a "45-degree angle neuron"), or additional population analyses such as a one-way ANOVA with angle difference as the main effect (or two-way ANOVA with angle difference as one of the main effect), would be very informative. Such data could offer valuable insights into how individual neurons contribute to the encoding of head-body angle differences-a detail that may also be reflected in the decoding results. Alternatively, it is possible that the encoding of head-body angle is inherently complex and only discernible via decoding methods applied to population activity. Either scenario would provide interesting and useful information to the field.

We have performed two additional analyses which are relevant to this comment. First, we attempted to relate the tuning for body and head orientation with the decoding of the head-body orientation angle. To do this, one needs to identify the neurons that contribute to the decoding of head-body orientation angles. For that, we employed a neuron-dropping analysis, similar to Chiang et al. (Chiang FK, Wallis JD, Rich EL. Cognitive strategies shift information from single neurons to populations in prefrontal cortex. Neuron. 2022 Feb 16;110(4):709-721.) to assess the positive (or negative) contribution of each neuron to the decoding performance. We performed cross-validated linear SVM decoding N times, each time leaving out a different neuron (using N-1 neurons; 2000 resamplings of pseudo-population vectors). We then ranked decoding accuracies from highest to lowest, identifying the ‘worst’ (rank 1) to ‘best’ (rank N) neurons. Next, we conducted N decodings, incrementally increasing the number of included neurons from 1 to N, starting with the worst-ranked neuron (rank 1) and sequentially adding the next (rank 2, rank 3, etc.). This analysis focused on zero versus straight angle decoding in the aSTS, as it yielded the highest accuracy. We applied it when training on MC and testing on HC for each pose. Plotting accuracy as a function of the number of included neurons suggested that less than half contributed positively to decoding (see Figure S3). We examined the tuning for head and body orientation of the 10 “best” neurons (Figure S3). For half or more of those the two-way ANOVA showed a significant interaction. These are indicated by the red color in the Figure. They showed a variety of tuning patterns for head and body orientation, suggesting that the decoding of the head-body orientation angle results from a combination of neurons with different tuning profiles.

Second, we have followed the suggestion of the reviewer to perform for each neuron of experiment 1 a one-way ANOVA with as factor head-body orientation angle. To do that, we combined all 64 trials that had the same head-body orientation angle. The percentage of neurons (required to be responsive in the tested condition) for which this one-way ANOVA was significant is shown in the Tables below for each region, separately for each pose (P1, P2), centering condition (MC = monkey-centered; HC = head-centered) and monkey subject (M1, M2). The percentages were low but larger than the expected 5% (Type 1 error), with a median of 16.5% (range: 3 to 23%) in aSTS and 8% for mSTS (range: 0-19%).

**Author response table 1. sa4table1:** 

aSTS															
MC								HC							
M1				M2				M1				M2			
P1		P2		P1		P2		P1		P2		P1		P2	
15%	n=34	9%	n=33	17%	n=35	23%	n=35	16%	n=32	3%	n=33	11%	n=36	17%	n=35

Interestingly, a higher percentage of the 10 best neurons for each pose (indicated by the star in the Figure above) showed a significant one-way ANOVA for angle (for P1, MC: 50% (95% confidence interval (CI): 19% – 81%); P1, HC: 70% (CI: 35% - 93%); P2, MC: 70% (CI: 35% – 93%); P2: HC: 50% (CI: 19%-81%)). These percentages were significantly higher than expected for a random sample from the population of neurons for each pose-centering combination (expected percentages listed in the same order as above: 16%, 13%, 16%, and 10%; all outside CI). Thus, for at least half of the “best” neurons, the response differed significantly among the head-orientation angles at the single neuron level. Nonetheless, the tuning profiles were quite diverse, suggesting population coding of head-body orientation angle. We have added this interesting and novel result to the Results (page 16) and Suppl. Material (Figure S3).

Minor comments:(1) Figure 4A, Fourth Row Example (Zero Angle vs. Straight Angle, Bottom of the P2 Examples): The order of the example stimuli might be incorrect- the 0{degree sign} head with 180{degree sign} body stimulus (leftmost) might be swapped with the 180{degree sign} head with 0{degree sign} body stimulus (5th from the left). While this ordering may be acceptable, please double-check whether it reflects the authors' intended arrangement.

We have changed the order of the two stimuli in Figure 4A, following the suggestion of the reviewer.

(2) Page 12, Lines 192-194: The text states, "Interestingly, some neurons (e.g. Figure 3D) were tuned to a particular combination of a head and body irrespective of centering." However, Figure 3D displays data for a total of 10 neurons. Could you please specify which of these neurons are being referred to in this context?

The wording was not optimal. We meant to say that some neurons were tuned to a particular combination of head and body orientation, like the third aSTS example neuron of Figure 3D. We have rephrased the sentence and clarified which example neuron we referred to.

(3) Page 28, Lines 470-471: The text states, "We observed no difference in response strength between anatomically possible and impossible configurations." Please clarify which data were compared for response strength, as I could not locate the corresponding analyses.

The anatomically possible and impossible configurations differ in the head-body orientation angle. However, as we reported before in the Results, there was no effect of head-body orientation angle on mean response strength across poses (Friedman ANOVA; all p-values for both poses and centerings > 0.1). We have clarified this now in the Discussion (page 28).

(4) Pages 40-43, Decoding Analyses: In experiments 2 and 3, were the decoding analyses performed on simultaneously recorded neurons? If so, such analyses might leverage trial-by-trial correlations and thus avoid confounds from trial-to-trial variability. In contrast, experiment 1, which used single-shank electrodes, would lack this temporal information. Please clarify how trial numbers were assigned to neurons in each experiment and how this assignment may have influenced the decoding performance.

For the decoding analyses of experiments 2 and 3, we combined data from different daily penetrations, with only units from the same penetration being recorded simultaneously. In the decoding analyses of each experiment, the trials were assigned randomly to the pseudo-population vectors, shuffling on each resampling the trial order per neuron. This shuffling abolishes noise correlations in the analysis of each experiment.

(5) Page 41, Lines 792-802: The authors state that "To assess the significance of the differences in classification scores between pairs of angles ... we computed the difference in classification score between the two pairs for each resampling and the percentile of 0 difference corresponded to the p-value." In a two-sided test under the null hypothesis of no difference between the distributions, the conventional approach would be to compute the p-value as the proportion of resampled differences that are as extreme or more extreme than the observed difference. Since a zero difference might be relatively rare, relying solely on its percentile could potentially misrepresent the tail probabilities relevant to a two-sided test. Could you clarify how their method addresses this issue?

This test is based on the computation of the distribution of the difference between classification accuracies across resamplings. This is similar to the computation of the confidence interval of a difference. Thus, we assess whether the theoretical zero value (=no difference; = null hypothesis) is outside the 2.5 and 97.5 percentile interval of the computed distribution of the empirically observed differences. We clarified now in the Methods (page 41) that for a two-tailed test the computed p-value (the percentile of the zero value) should be smaller than 0.025.

(6) Page 43, Lines 829-834: The manuscript explains: "The mean of 10 classification accuracies (i.e., of 10 resamplings) was employed to obtain a distribution (n=100) of the differences in classification accuracy ... The reported standard deviations of the classification accuracies are computed using also the means of 10 resamplings." I am unfamiliar with this type of analysis and am unclear about the rationale for calculating distributions and standard deviations based on the means of 10 resamplings rather than using the original distribution of classification accuracies. This resampling procedure appears to yield a narrower distribution and smaller standard deviations than the original data. Could you please justify this approach?

The logic of the analysis is to reduce the noise in the data, by averaging across 10 randomly selected resamplings, but still keeping a sufficient number of data (100 values) for a test.

**Reviewer #3 (Recommendations for the authors):**
(1) Some sentences are too long and difficult to parse. For example, in line 177: "the correlations between the responses to the 64 head-body orientation conditions of the two centerings for the neuron and pose combinations showing significant head-body interactions for the two centerings were similar to those observed for the whole population."

We have modified this sentence: For neuron and pose combinations with significant head-body interactions in both centerings, the correlations between responses to the 64 head-body orientation conditions were similar to those observed in the whole population.

(2) The authors argue in line 485: "in our study, a search bias cannot explain the body-inversion effect since we selected responsive units using both upright and inverted images." However, the body-selective patches were localized using upright images, correct?

The monkey-selective patches were localized using upright images indeed. However, we recorded in experiment 3 (and 2) also outside the localized patches (as we noted before in the Methods: “In experiments 2 and 3 we recorded from a wider region, which overlapped with the two monkey patches and the recording locations of experiment 1”). Furthermore, the preference for upright monkey images is not an all-or-nothing phenomenon: most units still responded to inverted monkeys. Also, we believe it is likely that the mean responses to the inverted bodies in the monkey patches, defined by upright bodies versus objects, would be larger than those to objects and we would be surprised to learn that there is a patch selective for inverted bodies that we would have missed with our localizer.

(3) Typo: line 447, "this independent"->"is independent"?

Corrected.